# LncRNA MIR200CHG inhibits EMT in gastric cancer by stabilizing miR-200c from target-directed miRNA degradation

Yixiao Zhu[1,2,3,6], Chengmei Huang[1,6], Chao Zhang[1,6], Yi Zhou[1,6], Enen Zhao[1], Yaxin Zhang[1], Xingyan Pan[1], Huilin Huang [1]✉, Wenting Liao [1]✉ & Xin Wang [2,4,5]✉

Gastric cancer (GC) is a heterogeneous disease, threatening millions of lives worldwide, yet the functional roles of long non-coding RNAs (lncRNAs) in different GC subtypes remain poorly characterized. Microsatellite stable (MSS)/epithelial-mesenchymal transition (EMT) GC is the most aggressive subtype associated with a poor prognosis. Here, we apply integrated network analysis to uncover lncRNA heterogeneity between GC subtypes, and identify MIR200CHG as a master regulator mediating EMT specifically in MSS/EMT GC. The expression of MIR200CHG is silenced in MSS/EMT GC by promoter hypermethylation, associated with poor prognosis. MIR200CHG reverses the mesenchymal identity of GC cells in vitro and inhibits metastasis in vivo. Mechanistically, MIR200CHG not only facilitates the biogenesis of its intronic miRNAs miR-200c and miR-141, but also protects miR-200c from target-directed miRNA degradation (TDMD) through direct binding to miR-200c. Our studies reveal a landscape of a subtype-specific lncRNA regulatory network, providing clinically relevant biological insights towards MSS/EMT GC.

Gastric cancer (GC) is the fifth most common malignancy and the fourth leading cause of cancer-related deaths, with over one million newly diagnosed cases each year worldwide[1]. Importantly, GC is a heterogeneous disease with complex genetic and epigenetic alterations, leading to diverse clinical manifestations, therapy responses, and prognoses[2]. Using one of the most widely used classifiers developed by the Asian Cancer Research Group (ACRG), GC can be subdivided into four molecular subtypes with distinct biological properties in relation to clinical outcomes[3]. Notably, patients of the MSS/EMT subtype showed a higher risk of recurrence and poorer overall survival. Despite the tremendous community effort on dissecting GC heterogeneity[4], the regulatory mechanism underlying the

MSS/EMT subtype remains elusive, hampering the design of novel targeted therapy for personalized treatment.

LncRNAs which are RNAs longer than 200 nucleotides without protein-coding potential, have drawn increasing attention for their critical roles in cancer initiation, progression, and metastasis[5–7]. Unlike miRNAs, which mainly silence their target mRNAs by interacting with the 3′-untranslated region (3′UTR) through seed sequences[8], lncRNAs exert their functions in epigenetic, transcriptional, or post-transcriptional regulation via more diverse mechanisms. In general, the regulatory mechanisms of lncRNAs in tumors can be divided into four types: signal, decoy, guide, and scaffold[9]. More importantly, the biological function of lncRNA is closely related to its subcellular

[1]State Key Laboratory of Oncology in South China, Guangdong Provincial Clinical Research Center for Cancer, Sun Yat-sen University Cancer Center, Guangzhou, Guangdong, China. [2]Department of Surgery, The Chinese University of Hong Kong, Hong Kong SAR, China. [3]National Clinical Research Centre for Geriatric Disorders, Department of Geriatrics, Xiangya Hospital, Central South University, Changsha, Hunan, China. [4]Li Ka Shing Institute of Health Sciences, Faculty of Medicine, The Chinese University of Hong Kong, Hong Kong SAR, China. [5]Shenzhen Research Institute, The Chinese University of Hong Kong, Shenzhen, Guangdong, China. [6]These authors contributed equally: Yixiao Zhu, Chengmei Huang, Chao Zhang, Yi Zhou. ✉e-mail: huanghl1@sysucc.org.cn; liaowt@sysucc.org.cn; xinwang@cuhk.edu.hk

localization[10]. Nuclear lncRNAs function in diverse nuclear events, including chromatin interactions and remodeling, transcriptional programs, and RNA processing[11], whereas cytoplasmic lncRNAs are widely involved in regulating mRNA stability and translation[12,13] and modulating protein posttranslational modifications[14]. The crosstalk between cytoplasmic lncRNAs and miRNAs can also have an impact on cancer development, metastasis, and orchestrate epithelial-mesenchymal plasticity[15]. Besides, some lncRNAs can serve as precursors of miRNAs to produce miRNAs, causing repression of target mRNAs[16]. Nevertheless, little was known about lncRNAs in (un-)stabilizing miRNAs. miRNAs could be induced to decay by extensive base-pairing with certain target mRNAs through a process termed TDMD, in which the extensive pairing can promote structural rearrangements of AGO2 for proteasomal degradation and expose the 3′ end of the miRNA to ribonucleases for degradation[17,18]. A few lncRNAs have been reported to direct miRNA degradation through TDMD. For example, the lncRNA Cyrano uses an extensively paired site to miR-7 to trigger destruction of miR-7[19]. The near-perfect miRNA binding site located in the lncRNA libra in zebrafish selectively triggers miR-29b destabilization through 3′ trimming[20]. Yet it remains elusive whether lncRNAs could stabilize miRNAs in TDMD.

Recent studies in GC have revealed a variety of lncRNAs to regulate GC migration, invasion, and metastasis[21]. For instance, lncRNA GMAN, which was upregulated in GC, was found to be associated with metastasis in patients and promoted the translation of EFNA1 by competitively binding to GMAN-AS[22]. LncRNA CA3-AS1 was reported to suppress GC migration and invasion by sponging miR-93-5p and targeting BTG3[23]. Despite the wealth of studies on individual lncRNAs, there is a lack of a systematic study to investigate the lncRNA regulatory network on a genome-wide scale. Moreover, as a highly heterogeneous disease entity, little is known about the subtype specificity of lncRNA functions in GC.

In the present study, we infer a lncRNA-mRNA regulatory network and identify MIR200CHG as a master regulator mediating the EMT pathway specifically in the MSS/EMT subtype of GC. We elucidate the biological functions and regulatory mechanisms of MIR200CHG in inhibiting EMT by stabilizing miR-200c from TDMD via comprehensive in silico analysis and experimental validations both in vitro and in vivo. Our work provides insights into the subtype specificity of lncRNAs in GC, and substantial evidence of MIR200CHG as a promising prognostic and predictive biomarker, as well as a potential therapeutic target for GC patients.

## Results

### The MSS/EMT GC subtype is characterized by distinct molecular properties with a unique lncRNA expression pattern

To investigate whether the four GC subtypes differ in biology, as reported previously[3], we performed comprehensive functional characterizations on three independent GC cohorts (TCGA, ACRG and GSE15459) with transcriptomic data. We found that the GC subtypes showed distinct biological properties consistently across all three cohorts (Supplementary Fig. 1a). More specifically, biological processes and signaling pathways related to EMT, transforming growth factor-β, and matrix remodeling were significantly upregulated in the MSS/EMT subtype (Supplementary Fig. 1a). The MSI subtype exhibited upregulated immune signatures and higher immune cell infiltration. Both the MSS/TP53- and MSS/TP53+ subtypes were epithelial and characterized by dysregulated metabolic pathways. Since the MSS/EMT subtype was previously found to be associated with the worst prognosis[3], we specifically focused on the EMT pathway in the following analysis. As expected, GSEA confirmed the significant upregulation of EMT in a subtype-specific manner (false discovery rate [FDR] = 0.01 by the permutation test, $n = 10,000$, Supplementary Fig. 1b). More specifically, representative EMT signature genes such as CDH11, VIM, ZEB1, and MMP2 were all significantly upregulated in the MSS/EMT

subtype across the three independent cohorts (all BH-adjusted $P < 0.0001$, Supplementary Fig. 1c).

Previous studies about the transcriptomic diversity of GC have been focused on mRNAs, leaving the lncRNA heterogeneity unexplored. Out of the total 14,656 lncRNAs annotated in the TCGA dataset, we identified 27 lncRNAs significantly differentially expressed (DE) between a specific GC subtype and the other subtypes (|log2FC| >1, BH-adjusted $P < 0.05$, Fig. 1a; Supplementary Table 1). Strikingly, most of these DE lncRNAs were specific to the MSS/EMT subtype (20 upregulated and two downregulated), revealing a very distinct lncRNA expression pattern in this subtype. Together, our in silico analysis confirmed the molecular characteristics associated with different GC subtypes consistently across multiple independent patient cohorts and revealed a distinct lncRNA expression pattern in the MSS/EMT subtype.

### Integrated network analysis identified MIR200CHG as a master regulator of the MSS/EMT subtype with strong prognostic and predictive values

To investigate the regulatory roles of lncRNAs in the MSS/EMT subtype of GC, we first inferred a lncRNA regulatory network using an integrative network analysis based on top DE mRNAs (Fig. 1b; Supplementary Data 1) and lncRNAs (Fig. 1c; Supplementary Table 2) in the TCGA dataset. As a result, a robust regulatory network with 1572 nodes (nine lncRNAs, 1563 mRNAs) and 2751 edges encoding lncRNA−mRNA associations was constructed using the 'ARACNE' algorithm[24] (*details in Methods*) (Fig. 1d). To further identify master regulatory lncRNAs of the EMT pathway in the MSS/EMT subtype, we performed a hypergeometric test for overrepresentation of EMT signature genes in the regulon of each lncRNA. As a result, three lncRNAs were prioritized: MIR200CHG, AC104083.1, and LINC00578 (Fig. 1d, e, hypergeometric test, BH-adjusted $P < 0.05$; Table 1). Among them, MIR200CHG (Fig. 2a) was strongly downregulated in the MSS/EMT subtype, while AC104083.1 (Fig. 2b) and LINC00578 (Fig. 2c) were significantly upregulated. Interestingly, most of the genes in the regulon of AC104083.1 (75.1%) or LINC00578 (80.3%) were predicted to be upregulated by the lncRNAs respectively, whereas the majority of the genes in the regulon of MIR200CHG (72.3%) were repressed by MIR200CHG (Fig. 1d). Using MIR200CHG as an example, we next sought to validate the robustness of the inferred regulon. More specifically, we investigated the expression of MIR200CHG regulon (332 genes) in the ACRG and GSE15459 datasets, respectively. As a result, the MIR200CHG regulon showed a consistent expression pattern across GC subtypes in the TCGA, ACRG and GSE15459 datasets (Supplementary Fig. 2a). Moreover, genes induced by MIR200CHG (92 genes) in the regulatory network derived from the TCGA dataset were more enriched in the non-MSS/EMT subtypes in both the ACRG and GSE15459 datasets (Supplementary Fig. 2b, d). On the contrary, genes repressed by MIR200CHG (240 genes) were more enriched in the MSS/EMT subtype in both the ACRG and GSE15459 datasets (Supplementary Fig. 2c, e).

We next sought to investigate the clinical relevance of the three master regulatory lncRNAs. Compared to early-stage GCs (T1), MIR200CHG was more lowly expressed in more advanced tumors (T2-T4) ($P = 0.039$, one-way ANOVA, Fig. 2d), whereas AC104083.1 and LINC00578 were significantly highly expressed ($P = 3.13 \times 10^{-4}$ and $8.55 \times 10^{-4}$, one-way ANOVA, Fig. 2e, f). A similar trend was also observed in patients with respect to the TNM stage (Supplementary Fig. 3a–c), indicating that these lncRNAs might be functionally important along GC progression. Given the worst prognosis previously observed in the MSS/EMT subtype[25], we next asked whether the three master regulatory lncRNAs were predictive of survival status. Univariate and multivariate Cox proportional hazard regression analysis revealed MIR200CHG as the only lncRNA with a significant prognostic value in the TCGA cohort (Fig. 2g;

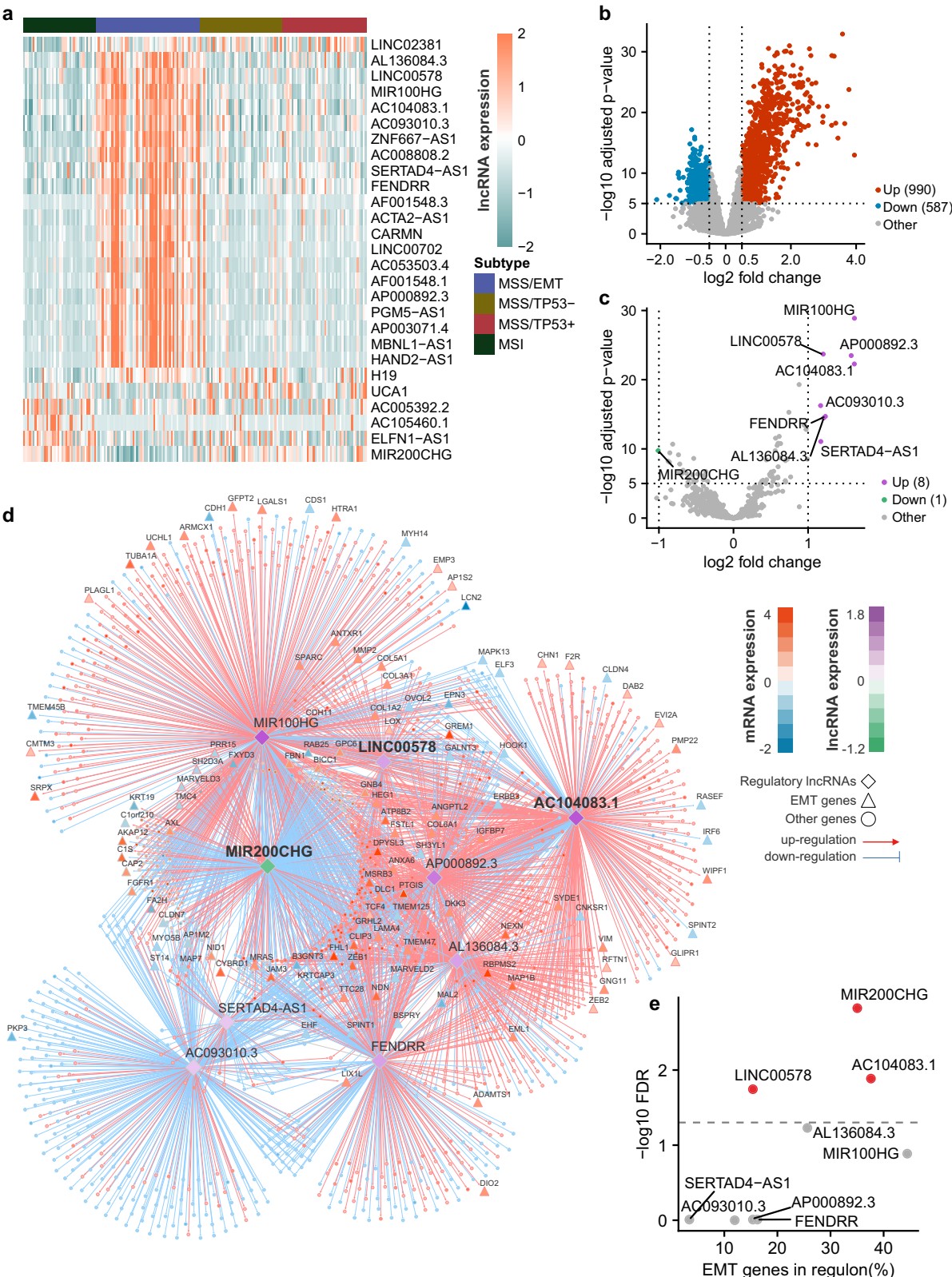

Supplementary Fig. 3d, e; Table 2). Kaplan–Meier survival analysis showed that patients with lower expression of MIR200CHG had much worse overall survival (hazard ratio = 0.45 [0.27–0.78], $P = 2.95 \times 10^{-3}$, log-rank test, Fig. 2h), indicating that MIR200CHG could serve as a prognostic biomarker for GC. Furthermore, to investigate whether MIR200CHG had any predictive value of drug sensitivity, we performed a drug sensitivity analysis of MIR200CHG

based on the CCLE drug dataset[26]. Interestingly, we found a significantly positive correlation between MIR200CHG expression and the IC50 value of irinotecan (a chemotherapy drug for colon and lung cancers) and lapatinib (a targeted therapy for HER2-positive breast cancer) (Supplementary Fig. 4a, b), suggesting the potential value of MIR200CHG expression to predict the sensitivities of the two drugs.

**Fig. 1 | Integrated network inference identified master regulatory lncRNAs of EMT specifically in the MSS/EMT subtype of GC. a** The heat map displays the expression profile of the differentially expressed lncRNAs in all four GC subtypes in the TCGA cohort ($n = 54$ samples for MSS/EMT, $n = 43$ samples for MSS/TP53-, $n = 44$ samples for MSS/TP53+, $n = 38$ samples for MSI). **b** The volcano plot of top differentially expressed mRNAs between the MSS/EMT subtype and the non-MSS/EMT subtypes in the TCGA cohort ($n = 179$ samples). **c** The volcano plot of top differentially expressed lncRNAs between the MSS/EMT subtype and the non-MSS/EMT subtypes in the TCGA cohort ($n = 179$ samples). **d** The lncRNA regulatory network inferred from an integrative analysis of mRNA and lncRNA expression profiles in the TCGA cohort. MIR200CHG, AC104083.1, and LINC00578 appeared as the most influential master regulatory lncRNAs for EMT

signature genes in GC. mRNAs and lncRNAs were colored according to the log2 fold change between the MSS/EMT subtype and the non-MSS/EMT subtypes of GC (orange: upregulation of mRNA expression; blue: downregulation of mRNA expression; purple: upregulation of lncRNA expression; green: downregulation of lncRNA expression). The edges between nodes were colored in red (induction) or blue (repression) based on the predicted regulatory relationship. EMT signature genes were denoted as triangles. **e** Statistical significance of the over-representation of a lncRNA's regulon for EMT signature genes vs. the proportion of EMT signature genes regulated by a lncRNA. LncRNAs with significant enrichment of EMT signature genes in the regulons were highlighted in red color. *P*-values were determined by moderated two-sided *t*-tests (**b**, **c**) and hypergeometric tests (**e**). *P*-values were adjusted for multiple testing (**b**, **c**, **e**).

## Table 1 | Master regulatory analysis result of lncRNAs in MSS/EMT GC

| Regulon | Universe size | Regulon size | Total hits | Expected hits | Observed hits | p-value | FDR |
|---|---|---|---|---|---|---|---|
| MIR200CHG | 1586 | 332 | 117 | 24.49 | 41 | 1.70E-04 | 1.50E-03 |
| AC104083.1 | 1586 | 414 | 117 | 30.54 | 44 | 3.00E-03 | 1.30E-02 |
| LINC00578 | 1586 | 132 | 117 | 9.74 | 18 | 6.10E-03 | 1.80E-02 |
| AL136084.3 | 1586 | 291 | 117 | 21.47 | 30 | 2.60E-02 | 5.90E-02 |
| MIR100HG | 1586 | 598 | 117 | 44.11 | 52 | 7.30E-02 | 1.30E-01 |
| AP000892.3 | 1586 | 262 | 117 | 19.33 | 18 | 6.70E-01 | 9.80E-01 |
| FENDRR | 1586 | 313 | 117 | 23.09 | 19 | 8.70E-01 | 9.80E-01 |
| SERTAD4-AS1 | 1586 | 69 | 117 | 5.09 | 4 | 7.60E-01 | 9.80E-01 |
| AC093010.3 | 1586 | 340 | 117 | 25.08 | 14 | 1.00E + 00 | 1.00E + 00 |

*FDR* false discovery rate.

## The hypermethylation of MIR200CHG promoter in the GC MSS/EMT subtype underlies its low expression

To further verify the repression of MIR200CHG in the MSS/EMT subtype of GC, we analyzed two other independent cohorts (ACRG and GSE15459) and cell lines (CCLE). Remarkably, MIR200CHG was significantly downregulated in the MSS/EMT subtype in both the ACRG dataset (Fig. 3a) and the GSE15459 dataset (Fig. 3b). More strikingly, in the CCLE dataset, no expression of MIR200CHG was observed in any of the six MSS/EMT GC cell lines (Fig. 3c; Supplementary Fig. 4c). RT-qPCR was conducted to validate the repressed expression of MIR200CHG in three representative MSS/EMT cell lines, namely, SNU668, MKN1, and Hs746T. As expected, little expression of MIR200CHG was observed in the three MSS/EMT cell lines (Fig. 3d), firmly demonstrating the repression of MIR200CHG in the MSS/EMT subtype.

Since promoter methylation is a frequent epigenetic mechanism for gene silencing, we postulated that the suppression of MIR200CHG in the MSS/EMT subtype may be attributed to hypermethylation. To test this hypothesis, we compared the methylation levels of the MIR200CHG promoter between GC subtypes in the TCGA tissue samples and the CCLE cell lines. As expected, we found that the MIR200CHG promoter had significantly higher methylation levels in the MSS/EMT subtype than in the non-MSS/EMT subtypes in the TCGA dataset (Fig. 3e). Moreover, the methylation level of the MIR200CHG promoter was significantly inversely correlated with the expression level of MIR200CHG (PCCs = −0.42, $P = 4.30 \times 10^{-8}$, Fig. 3e). Similarly, we also observed the dramatic hypermethylation of the MIR200CHG promoter in MSS/EMT cell lines (Supplementary Fig. 4d). Furthermore, the de-methylation treatment of two MSS/EMT cell lines – SNU668 and Hs746T – by 5-Aza (5-aza-2′-deoxycytidine, a DNA methyltransferase inhibitor) resulted in the re-expression of MIR200CHG (Fig. 3f) and subsequent significant decrease of cell migration and invasion (Fig. 3g, h), suggesting the demethylation of MIR200CHG inhibits the mesenchymal identity of MSS/EMT cells.

Given the subtype-specific expression and methylation patterns of MIR200CHG, we investigated the discriminative power of MIR200CHG for the identification of the MSS/EMT GC. The results in the TCGA cohort showed that MIR200CHG promoter methylation had stronger predictive power (area under the curve [AUC] = 0.87) than MIR200CHG expression in the TCGA (AUC = 0.82), ACRG (AUC = 0.78), and GSE15459 (AUC = 0.79) cohorts (Fig. 3i).

Together, our results suggest that the elevated methylation of the MIR200CHG promoter is responsible for the inhibition of MIR200CHG in MSS/EMT GC, providing an indicator of the mesenchymal identity of GC and a potential therapeutic strategy to rescue the phenotype of MIR200CHG dysregulation.

## Enforced MIR200CHG expression reversed the mesenchymal identity of GC cells in vitro and inhibited LNs metastasis in vivo

We next investigated the biological effects of MIR200CHG in vitro. The functional modes of lncRNAs depend largely on their cellular location[9]. The coding potential assessment tool[27] predicted that MIR200CHG has little coding potential and is a genuine non-coding RNA (Supplementary Fig. 4e). Subsequently, we used RT-qPCR and RNA FISH to assess the subcellular localization of MIR200CHG in intact cells. The results showed that more than 85% of the MIR200CHG transcripts in NCI-N87 cells were localized in the cytoplasm (Fig. 3j; Supplementary Fig. 5a), suggesting that MIR200CHG functions at the post-transcriptional level.

To investigate whether MIR200CHG affects EMT in GC, we established stable MIR200CHG overexpression cells in two MSS/EMT cell lines – Hs746T and SNU668 (Supplementary Fig. 5b). We next evaluated the functional and morphological changes of the treated GC cells and detected the molecular indicators of EMT. Transwell analysis and wound healing analysis revealed that enforced expression of MIR200CHG significantly hampered the migration ability of Hs746T and SNU668 cells compared with the vector control cells (Fig. 4a, b). F-actin staining showed that Hs746T and SNU668 cells with ectopic

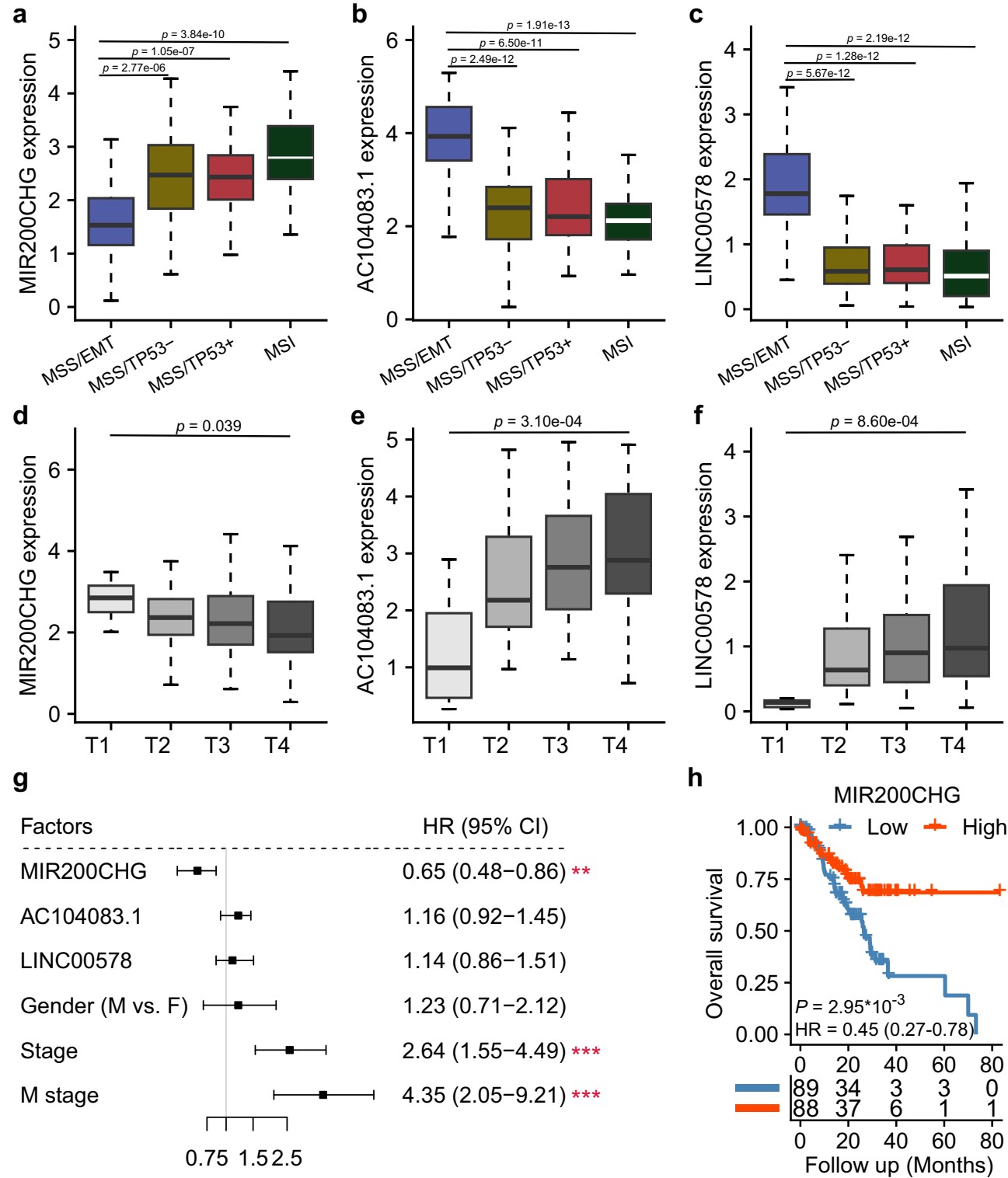

**Fig. 2 | Clinical associations of the master regulatory lncRNAs.** Boxplots show significant differential expression of MIR200CHG (**a**), AC104083.1 (**b**), and LINC00578 (**c**) between the MSS/EMT subtype and the non-MSS/EMT subtypes in the TCGA cohort ($n = 54$ samples for MSS/EMT, $n = 43$ samples for MSS/TP53-, $n = 44$ samples for MSS/TP53+, $n = 38$ samples for MSI). P-values were based on two-sided Wilcoxon rank-sum tests. Boxplots show significant associations between the expression of MIR200CHG (**d**), AC104083.1 (**e**), and LINC00578 (**f**) with tumor (T) stage ($n = 8$ samples for T1, $n = 35$ samples for T2, $n = 94$ samples for T3,

$n = 41$ samples for T4). P-values were based on One-way ANOVA. **g** The univariate Cox regression analysis of lncRNAs and typical clinical factors in the TCGA cohort ($n = 177$ samples, Wald tests, **(MIR200CHG) $P = 0.0031$, ***(Stage) $P = 0.00036$, ***(M stage) $P = 0.00012$. **h** Patients with lower MIR200CHG expression had significantly worse overall survival ($n = 177$ samples, log-rank test). Boxes in all boxplots extend from the 25th to the 75th percentile and the lines indicate the median. The whiskers are drawn to the 5th and the 95th percentile.

**Table 2 | Multivariate Cox analysis result**

| Features | beta | HR | HR (95% CI) | p-value |
|---|---|---|---|---|
| Gender: M vs. F | 0.065 | 1.07 | 1.07 (0.61–1.87) | 0.82 |
| T stage: T34 vs. T12 | 0.83 | 2.30 | 2.30 (1.31–4.02) | 0.0035 |
| M stage: M1 vs. M0 | 0.93 | 2.53 | 2.53 (1.13–5.64) | 0.024 |
| MIR200CHG | −0.41 | 0.66 | 0.66 (0.48–0.93) | 0.016 |
| AC104083.1 | 0.051 | 1.05 | 1.05 (0.74–1.50) | 0.78 |
| LINC00578 | −0.15 | 0.86 | 0.86 (0.52–1.40) | 0.54 |

MIR200CHG overexpression induced epithelial cell-like morphological features, such as less formation of pseudo-feet, and fewer stress fiber actin bundles (Fig. 4c). Furthermore, MIR200CHG overexpression led to a significant increase in the expression of epithelial marker ZO-1, and a reduction in the expression of mesenchymal markers, including fibronectin, ZEB1, and vimentin (Fig. 4c–e).

LNs are the most common metastatic sites of GC. To further examine the effect of MIR200CHG on LN metastasis in vivo, we established a mouse inguinal lymphatic metastasis model. Hs746T cells overexpressed with MIR200CHG or vector were implanted in the footpads of nude mice ($n = 8$). The lymph nodes were dissected, and metastasis was observed under the microscope. The volume of inguinal LNs in the vector group was significantly larger than that in the MIR200CHG-overexpressing group (Fig. 4f, g). In addition, seven mice in the vector control group formed LN metastasis. However, only four mice in the MIR200CHG-overexpressing group formed LN metastasis (Fig. 4g). Haematoxylin and eosin (HE) staining of LNs and IHC of panCK showed that overexpression of MIR200CHG significantly inhibited the ability of GC cells to metastasize to the LNs (Fig. 4h). IHC analyses showed that overexpression of MIR200CHG significantly upregulated E-cadherin and downregulated vimentin in primary tumor and LN metastasis in vivo (Fig. 4h; Supplementary Fig. 5d).

## MIR200CHG inhibition induced the EMT identity of GC cells in vitro and promoted metastasis in vivo

Next, we established MIR200CHG knockdown cells in two non-MSS/EMT cell lines—NUGC4 and NCI-N87 (Supplementary Fig. 5c), and examined the effect of MIR200CHG inhibition on EMT in vitro. As expected, the knockdown of MIR200CHG induced a reverse change from epithelial morphology to mesenchymal morphology and significantly promoted the migration of NUGC4 and NCI-N87 cells in vitro (Fig. 5a, b; Supplementary Fig. 6a). In addition, knockdown of MIR200CHG increased formation of pseudo-foot in NUGC4 and NCI-N87 cells, accompanied by decreased expression of the epithelial markers ZO-1 and E-cadherin and increased expression of the mesenchymal marker vimentin, as revealed by immunofluorescence analysis (Fig. 5c). Moreover, MIR200CHG knockdown decreased the expression of E-cadherin and ZO-1 and increased the expression of fibronectin and ZEB1, as examined by RT-qPCR and Western blotting (Supplementary Fig. 6b, c).

Furthermore, the mouse inguinal lymphatic metastasis models with NCI-N87 scramble and MIR200CHG-knockdown cell lines were established to examine the effects of MIR200CHG silencing on LN metastasis ($n = 6$). As shown in Fig. 5d, e, the volume of inguinal LNs in the knockdown group was significantly larger than that in the scramble group. In addition, only half of the mice in the scramble control group formed LN metastasis. However, all mice in the MIR200CHG-knockdown group formed LN metastasis (Fig. 5e, left). Moreover, IHC staining showed that knockdown of MIR200CHG restored the expression of vimentin and decreased the expression of E-cadherin in both primary tumors and LN metastasis (Fig. 5f; Supplementary Fig. 6d). To further confirm the role of MIR200CHG in GC metastasis, we have established a mouse peritoneal metastasis model. Our data

showed that inhibition of MIR200CHG aggravated peritoneal metastasis (Fig. 5h) and decreased overall survival rate in mice (Fig. 5i). Inhibition of MIR200CHG led to faster weight gain in mice, reflecting the rate of ascites production (Fig. 5j). Together, these results suggest that MIR200CHG is essential for inhibiting GC metastasis.

## The transcription of *MIR200CHG* facilitates the biogenesis of miR-200c and miR-141

Evaluation of the genomic localization of *MIR200CHG* revealed that the *MIR200CHG* gene was the host gene for two well-studied miRNAs – miR-200c and miR-141 – which are located in the intron region of the *MIR200CHG* gene as intronic miRNAs and share the same promoter with *MIR200CHG*, as indicated by the GeneCards database[28] (Supplementary Fig. 7a). It has been reported that intronic miRNAs often co-express with their host genes and function in a synergetic or antagonistic way in cancers[29]. Indeed, we found a strong positive correlation between MIR200CHG expression and miR-200c and miR-141 expression in the TCGA cohort (Fig. 6a; Supplementary Fig. 7b). Consistent with MIR200CHG, we observed low expression of miR-200c and miR-141 in the three MSS/EMT cell lines but high expression in five non-MSS/EMT cell lines by RT-qPCR (Supplementary Fig. 7c, d). Furthermore, the de-methylation treatment of SNU668 and Hs746T also resulted in the re-expression of miR-200c and miR-141 (Supplementary Fig. 7e), which were parallel to MIR200CHG (Fig. 3f). These data suggested that miR-200c and miR-141 are intronic miRNAs using the same transcriptional start sites of *MIR200CHG* to initiate transcription, demonstrating the transcription of *MIR200CHG* facilitates the biogenesis of miR-200c and miR-141.

To investigate to which extent the MIR200CHG regulon overlaps with the predicted targets of miR-200c or miR-141, we further performed similar network inference based on GC mRNA and miRNA expression profiles. Interestingly, we found there were 714 and 845 genes in the miR-200c and miR-141 regulons, respectively, with significant overlaps with the MIR200CHG regulon ($P = 1.15 \times 10^{-17}$ and $P = 5.38 \times 10^{-16}$, hypergeometric tests, Supplementary Fig. 7f, g), suggesting that MIR200CHG may be functionally relevant to miR-200c and miR-141.

## MIR200CHG stabilizes miR-200c by directly binding to miR-200c

We further investigated the functional relevance and were surprised to find that MIR200CHG could regulate miR-200c and miR-141 at the post-transcriptional level. More specifically, RT-qPCR revealed that the expression of miR-200c and miR-141 were decreased by MIR200CHG knockdown and rescued by MIR200CHG overexpression (Fig. 6b, Supplementary Fig. 7h). Moreover, MIR200CHG overexpression significantly increased, while MIR200CHG knockdown decreased the half-life of miR-200c but not miR-141 (Fig. 6c, Supplementary Fig. 7i). This data indicated that miR-200c, while not miR-141, is a predominant target of MIR200CHG at the post-transcriptional level in GC cells. Since cytoplasmic-enriched lncRNAs could regulate target genes expression at the post-translational level through the direct interaction with RNAs or proteins[30], we subsequently investigated the potential interactions between MIR200CHG and miR-200c. MIR200CHG-MS2 was overexpressed in HEK293T cells and MS2 RIP was performed to pull down endogenous RNAs and proteins associated with MIR200CHG. RT-qPCR showed that miR-200c but not miR-141, was significantly associated with MIR200CHG (Fig. 6d, e; Supplementary Fig. 7j). To further confirm this result, miR-200c or miR-141 was overexpressed in HEK293T cells and RIP was performed using an anti-AGO2 antibody to pull down RNAs binding to miR-200c or miR-141. RT-qPCR showed that MIR200CHG and miR-200c were specifically enriched in an anti-AGO2 antibody-associated complex (Fig. 6f).

Using the LncTar database[31], we then predicted the potential miR-200c binding sites of MIR200CHG, and consequential pairing was

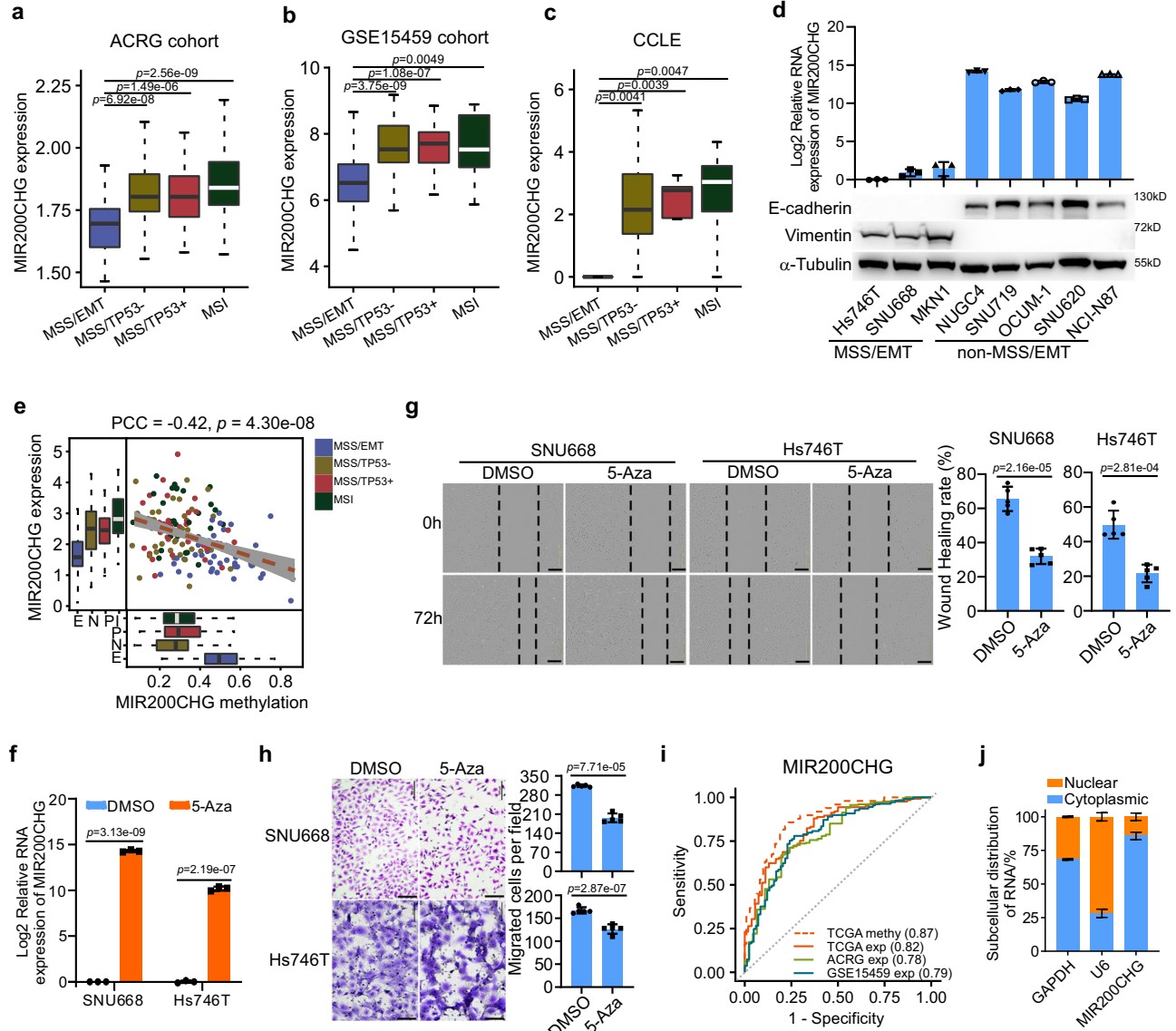

**Fig. 3 | Validation of MIR200CHG differential expression and its upstream regulatory mechanism.** MIR200CHG expression was significantly lower in the MSS/EMT subtype of GC in the ACRG cohort (**a**, n = 46 samples for MSS/EMT, n = 107 samples for MSS/TP53-, n = 79 samples for MSS/TP53 + , n = 68 samples for MSI), the GSE15459 cohort (**b**, n = 83 samples for MSS/EMT, n = 61 samples for MSS/TP53-, n = 37 samples for MSS/TP53 + , n = 11 samples for MSI), and the CCLE GC cell lines (**c**, n = 6 cell lines for MSS/EMT, n = 18 cell lines for MSS/TP53-, n = 5 cell lines for MSS/TP53+, n = 8 cell lines for MSI). *P*-values were based on two-sided Wilcoxon rank-sum tests. **d** Real-time PCR analysis of MIR200CHG expression and western blots show the differential expression of E-cadherin and vimentin in the MSS/EMT and non-MSS/EMT subtypes of GC cell lines. **e** The scatter plot shows the inverse correlation between MIR200CHG promoter methylation and MIR200CHG expression in different GC subtypes in the TCGA cohort (n = 49 samples for E, n = 40 samples for N, n = 40 samples for P, n = 31 samples for I). E: MSS/EMT, N: MSS/TP53-, P: MSS/TP53+, I: MSI. PCC: Pearson's correlation coefficient. *P*-value was based on two-sided Pearson's correlation test. **f** 5-Aza (0.5 μM) treatment of the

SNU668 and Hs746T cell lines revealed that a reduction in MIR200CHG promoter methylation resulted in MIR200CHG re-expression. **g** Wound healing analysis of SNU668 and Hs746T cells treated with 5-AZA or DMSO. Scale bar, 200 μm. **h** Transwell chamber analysis of Hs746T and SNU668 cells treated with 5-AZA or DMSO. Scale bar, 50 μm. **i** Receiver operating characteristic (ROC) curves illustrate the performance to employ MIR200CHG promoter methylation or expression to predict the MSS/EMT subtype in the TCGA, ACRG, and GSE15459 cohorts. **j** Stacked bars show the proportion of MIR200CHG localized in the cytoplasm and nuclear of NCI-N87 cells. GAPDH served as the cytoplasmic internal control. U6 served as the nuclear internal control. Boxes in all box-plots extend from the 25th to the 75th percentile and the lines indicate the median. The whiskers are drawn to the 5th and the 95th percentile. Each bar in bar plots represents the mean ± standard deviation of three biologically independent samples (**d, f, j**) and five biologically independent samples (**g, h**). *P*-values were determined by two-sided Student's *t*-tests (**f–h**). Source data are provided as a Source Data file.

shown (Supplementary Fig. 7k). To further determine whether MIR200CHG directly binds to miR-200c, biotin-labeled full-length MIR200CHG probes (M-HL) and biotin-labeled RNA oligos containing predicted miR-200c binding sites (MIR-S) (Supplementary Fig. 7k) were synthesized and co-incubated with total RNA from NCI-N87 cell lysates. RNA pulldown followed by RT-qPCR showed that full-length MIR200CHG bound to endogenous miR-200c but not miR-141

(Supplementary Fig. 7l). The interaction of endogenous miR-200c and MIR-S probe confirmed the reliability of predicted miR-200c binding sites of MIR200CHG (Fig. 6g). The RNA-RNA pulldown experiment followed by RT-qPCR further showed the interaction of miR-200c mimics with MIR-S probes, demonstrating that miR-200c is a direct target of MIR200CHG (Fig. 6h). Furthermore, we constructed MIR200CHG mutants with mutations of predicted miR-200c binding

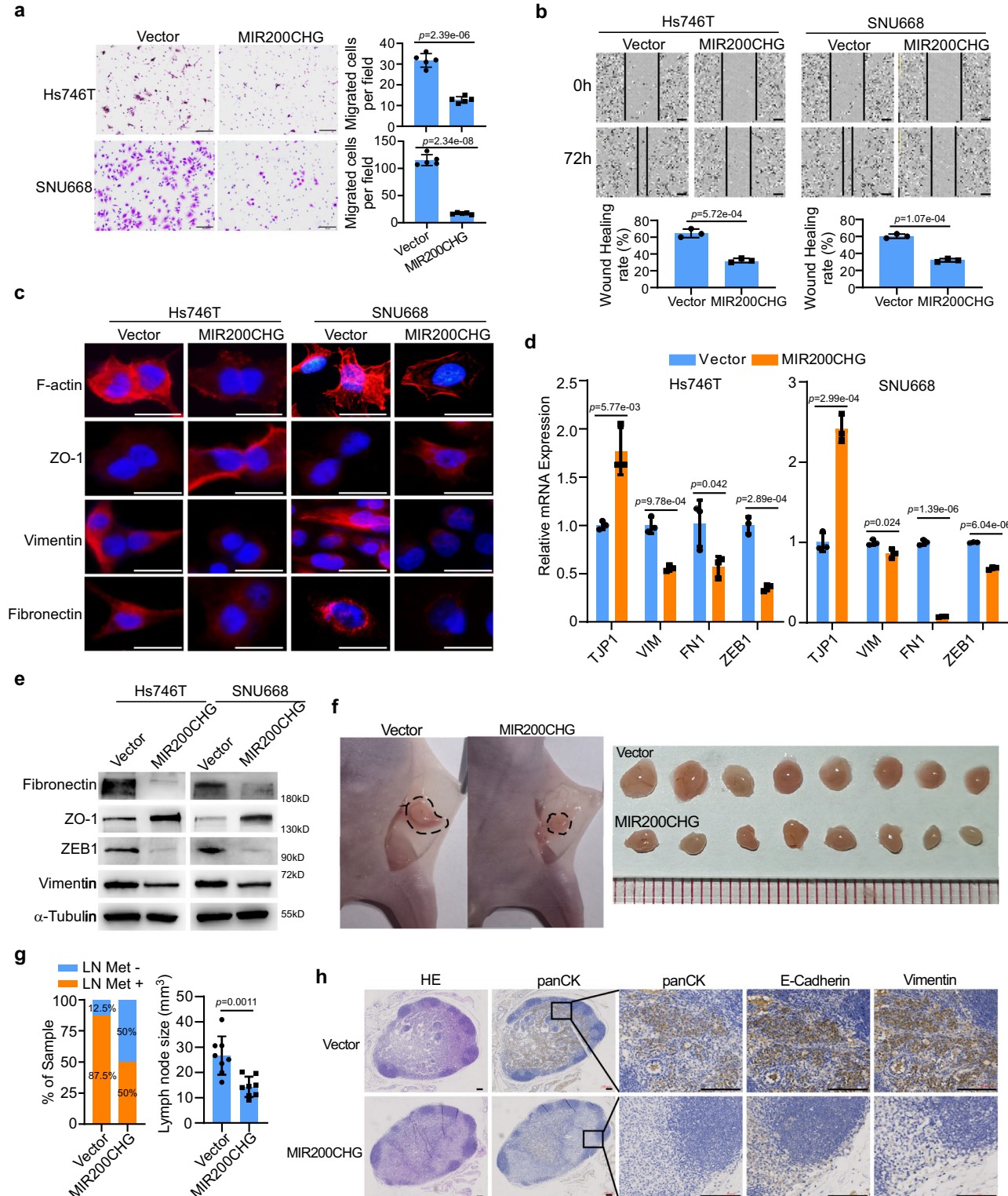

**Fig. 4 | Enforced MIR200CHG expression reversed the mesenchymal identity of GC cells in vitro and inhibited lymph node metastasis in vivo. a** Transwell chamber analysis of Hs746T and SNU668 cells. Each bar represents the mean ± standard deviation of five biologically independent samples. Scale bar, 200 μm. **b** Wound healing analysis of Hs746T and SNU668 cells for 72 h. Each bar represents the mean ± standard deviation of three biologically independent samples. Scale bar, 200 μm. **c** Immunofluorescence staining of ZO-1, Vimentin, Fibronectin, and F-actin in the indicated cells. Nuclei were counterstained with DAPI. Scale bar, 20 μm. **d** Expression of EMT relevant markers as determined by RT-qPCR. Each bar represents the mean ± standard deviation of three biologically independent samples. **e** The protein expression of ZO-1, Vimentin, Fibronectin, and ZEB1 in the

indicated cells, as assessed by western blotting. **f** Representative images of the mouse inguinal lymphatic metastasis models established with Hs746T vector-expressing and MIR200CHG-overexpressing cell lines (*n* = 8 mice). **g** Percentages of lymph node metastasis status (left) and lymph node size (right) in all groups. Each bar represents the mean ± standard deviation of eight biologically independent samples. **h** Haematoxylin and eosin staining and immunocytochemical analysis of panCK, E-cadherin, and Vimentin in the lymph node metastatic tumor. Scale bar, 100 μm. The experiments were repeated three times independently with similar results (**c**, **e**, **h**). *P*-values were determined by two-sided Student's *t*-tests (**a**, **b**, **d**, **g**). Source data are provided as a Source Data file.

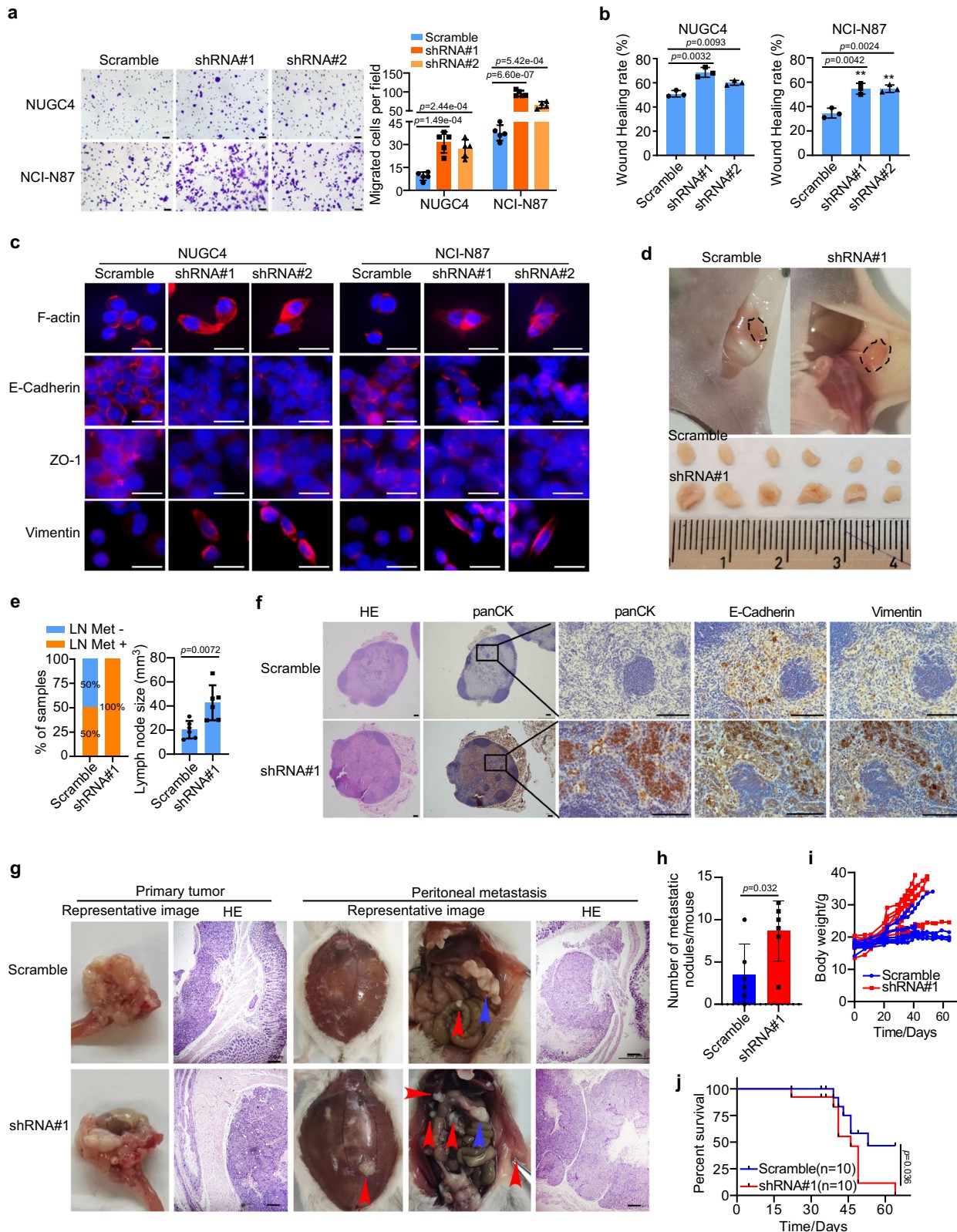

site (Mut-MS2), and observed that mutations completely abolished the interaction between MIR200CHG and miR-200c (Fig. 6i). RNA stability assay showed that enforced expression of wild-type MIR200CHG, but not mutant MIR200CHG, significantly stabilized miR-200c (Fig. 6j). Previous studies have shown that AGO2 is the core component of RNA-induced silencing complex (RISC) and regulates mRNA degradation. Through FISH-IF experiments, we found partial co-localization

between MIR200CHG and AGO2, as well as between MIR200CHG and DICER - another marker for RISC (Supplementary Fig. 7m). These results suggest that MIR200CHG stabilizes miR-200c in the RISC. Altogether, these data demonstrated that MIR200CHG is directly binding to miR-200c and could stabilize miR-200c.

The miR-200 family, including miR-200c, miR-141, miR-429, miR-200a and miR-200b, controls the expression of many genes that play

**Fig. 5 | MIR200CHG inhibition induced the EMT identity of GC cells in vitro and promoted metastasis in vivo. a** The transwell chamber analysis of NUGC4 and NCI-N87 cells. Each bar represents the mean ± standard deviation of five biologically independent samples. Scale bar, 50 μm. **b** Wound healing analysis of NUGC4 and NCI-N87 cells for 72 h. Each bar represents the mean ± standard deviation of three biologically independent samples. **c** Immunofluorescence of ZO-1, E-cadherin, Vimentin, and F-actin in NUGC4 and NCI-N87 cells. Nuclei were counterstained with DAPI. Scale bar, 20 μm. **d** Representative images of the mouse inguinal lymphatic metastasis models established with NCI-N87 Scramble and MIR200CHG-knockdown cell lines (*n* = 6 mice). **e** Percentages of lymph node metastasis status (left) and lymph node size (right) in all groups. Each bar represents the mean ± standard deviation of six biologically independent samples. **f** Haematoxylin and

eosin staining and immunocytochemical analysis of panCK, E-cadherin, and vimentin in the lymph node metastatic tumor. Scale bar, 100 μm. **g** Representative images of the mouse primary gastric tumor and peritoneal metastasis established with NCI-N87 Scramble and MIR200CHG-knockdown cell lines. **h** The number of peritoneal metastasis nodules of NSG mice. Each bar represents the mean ± standard deviation of six biologically independent samples. **i** Body weight of NSG mice. **j** The survival of NSG mice with NCI-N87 Scramble and MIR200CHG-knockdown tumors (*n* = 10). The experiments were repeated three times independently with similar results (**c**, **f**). *P*-values were determined by two-sided Student's *t*-tests (**a**, **b**, **e**, **h**) and a log-rank test (**j**). Source data are provided as a Source Data file.

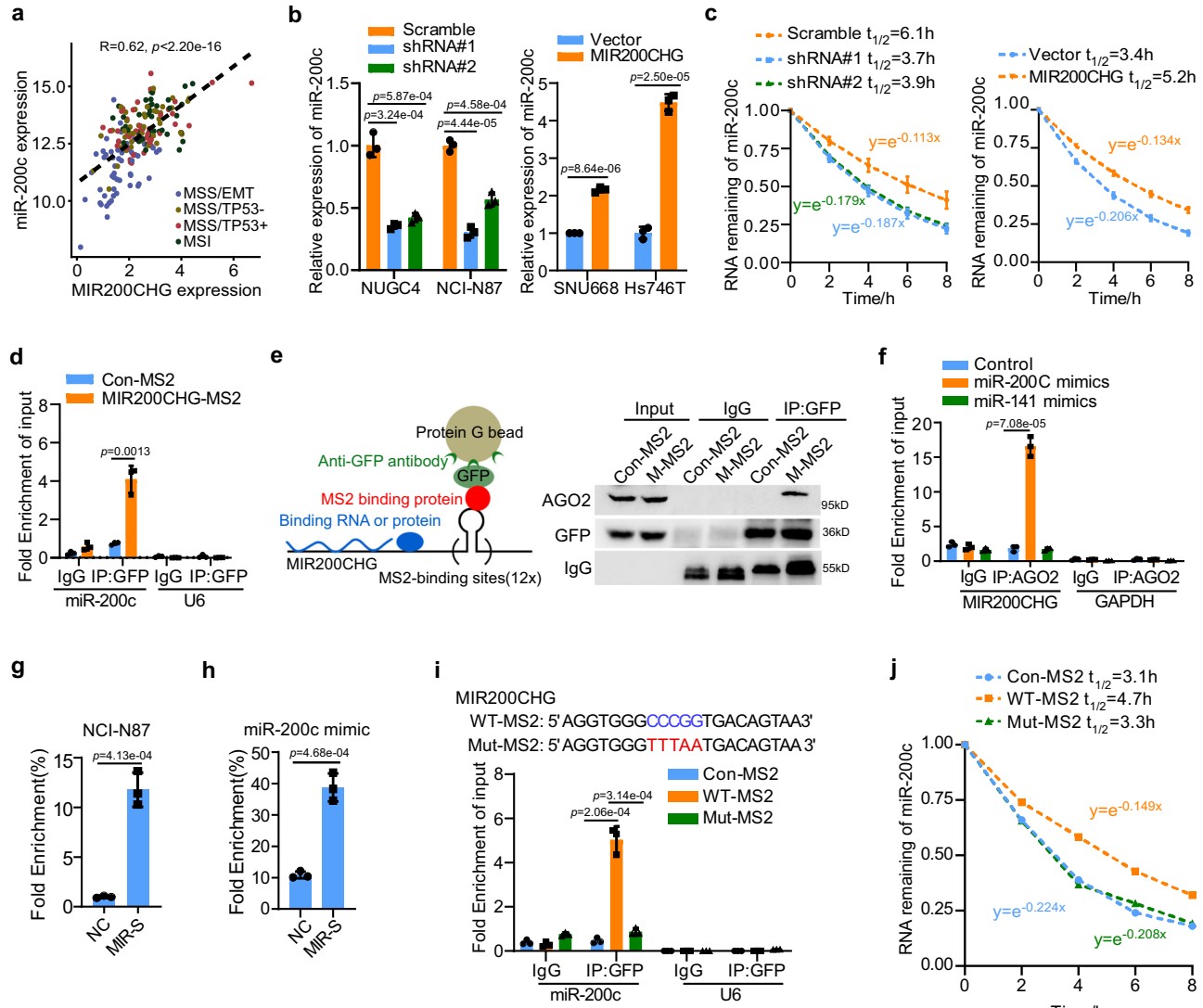

**Fig. 6 | MIR200CHG protects miR-200c from TDMD by binding to miR-200c-bound AGO2. a** The scatter plot shows a positive correlation between MIR200CHG expression and miR-200c expression in the TCGA cohort (*n* = 54 samples for MSS/EMT, *n* = 43 samples for MSS/TP53-, *n* = 44 samples for MSS/TP53+, *n* = 38 samples for MSI). *P*-value was based on a two-sided Pearson's correlation test. **b** miR-200c expression in MIR200CHG-knockdown (left) and MIR200CHG-overexpressing cell lines (right) was determined by RT-qPCR. **c** The reduced half-life of miR-200c by silencing MIR200CHG (left) and extended half-life of miR-200c by overexpressing MIR200CHG (right). Cells were treated with 5 μmol/L actinomycin D. **d** MS2 RIP and RT-qPCR analyses show the interaction of MIR200CHG with miR-200c in HEK293T cells. **e** The schematic diagram (left) and western blot (right) of MS2 RIP assay. The experiment was repeated three times independently with similar results. **f** The AGO2 RIP assay showed that both MIR200CHG and miR-200c were bound to

AGO2. **g** The RNA-pull down and RT-qPCR showing the association of miR-200c with MIR200CHG in NCI-N87 cells. **h** The RNA-RNA pull down and RT-qPCR showing the direct binding of MIR200CHG and miR-200c. **i** The corresponding mutant form (Mut-MS2) with the predicted miR-200c binding site mutated is shown (top). HEK293T cells were transfected with negative control (Con-MS2), vectors containing wild-type (WT-MS2) or mutated MIR200CHG (Mut-MS2) followed by MS2-RIP assay. RT-qPCR analysis showed the interaction of MIR200CHG with miR-200c in HEK293T cells transfected with Con-MS2, WT-MS2 or mutated Mut-MS2. **j** The RT-qPCR analysis showed the extending miR-200c half-life by overexpressing wild-type MIR200CHG but not mutant MIR200CHG. Each bar in bar plots represents the mean ± standard deviation of three biologically independent samples (**b**, **d**, **f–i**). *P*-values were determined by two-sided Student's *t*-tests (**b–d**, **f–i**). Source data are provided as a Source Data file.

important roles in EMT[32]. Therefore, we further analyzed whether other members of the miR-200 family are involved in the regulation of MIR200CHG. In the TCGA and CCLE datasets, miR-429, miR-200a and miR-200b showed significantly lower expression in the MSS/EMT subtype (Supplementary Fig. 8a). We used lncTAR and miRanda to simultaneously predict the binding sites of MIR200CHG and miR-200 family. The results showed that MIR200CHG required the lowest free energy to bind to miR-200c and miR-429 (Supplementary Fig. 8b, c). MS2-RIP results demonstrated that miR-200c and miR-429 were significantly associated with MIR200CHG (Supplementary Fig. 8d). These results suggested that similar to miR-200c, miR-429 can be bound and regulated by MIR200CHG, but the specific molecular mechanisms need to be further explored.

### MIR200CHG protects miR-200c from TDMD by inhibiting AGO2 degradation

Our data demonstrated that MIR200CHG stabilizes miR-200c, but the underlying mechanism remains unclear. miRNAs could be induced to decay by extensive base-pairing with certain target mRNAs, which is known as TDMD[17]. Given that MIR200CHG directly interacted with miR-200c, we speculated that MIR200CHG could protect miR-200c by suppressing TDMD-mediated miR-200c degradation. To address this question, we chose ZEB1, a well-known target of miR-200c, for further study[33]. We found a significantly negative correlation between MIR200CHG expression and ZEB1 expression in the TCGA cohort (Supplementary Fig. 9a). In addition, MS2 RIP showed that MIR200CHG associated with the ZEB1 mRNA (Supplementary Fig. 9b) and the half-life of the ZEB1 mRNA was increased in the absence of MIR200CHG and decreased with MIR200CHG overexpression (Supplementary Fig. 9c). ZEB1 knockdown also increased the half-life of miR-200c (Supplementary Fig. 9d). More interestingly, we found that the MIR200CHG binding site at miR-200c did not overlap with the seed region of miR-200c (Fig. 7a). However, the overexpression of MIR200CHG significantly inhibited the interaction between ZEB1 and miR-200c in vitro and in vivo (Fig. 7b, c; Supplementary Fig. 9e, f), suggesting the binding of MIR200CHG in the non-seed region of miR-200c inhibited the extending pairing of miR-200c and ZEB1 mRNA. In addition, a potential MIR200CHG binding site of miR-141 was predicted in the LncTar database. However, it was partially overlapped with the seed sequence of miR-141, which may suggest a competitive binding way, while not TDMD mechanism, of miR-141 between MIR200CHG and the targeted mRNAs of miR-141 (Supplementary Fig. 9g).

Additionally, we found that MIR200CHG is associated with AGO2 protein by RNA pulldown in NCI-N87 cells (Supplementary Fig. 9h). Inhibition of miR-200c impaired the binding of AGO2 to MIR200CHG (Supplementary Fig. 9i). Furthermore, mutations of the predicted miR-200c binding site (Mut-MS2) also abolished the interaction between MIR200CHG and AGO2 (Supplementary Fig. 9j), suggesting that MIR200CHG is involved in miR-200c-bound AGO2. We then investigated whether MIR200CHG could stabilize miR-200c by suppressing the proteasomal degradation of AGO2. The results showed that knockdown or overexpression of MIR200CHG resulted in the corresponding decrease or increase in AGO2 expression (Fig. 7d). In addition, the knockdown of MIR200CHG dramatically shortened the half-life of AGO2 protein (Fig. 7e), while overexpression of MIR200CHG prolonged the half-life of AGO2 protein but was reversed after the mutation of the miR-200c binding site (Fig. 7f). Moreover, AGO2 expression was partially rescued after adding MG132 (a proteasome inhibitor), which indicates that MIR200CHG is involved in the proteasome degradation pathway of AGO2 (Fig. 7g, h). To further validate this finding, we evaluated AGO2 polyubiquitination in MIR200CHG-overexpressing and MIR200CHG-silenced GC cells. As expected, AGO2 ubiquitination was substantially decreased when MIR200CHG was overexpressed, which was largely impaired by the mutation in the miR-200c binding sites (Fig. 7i). On the contrary, AGO2 ubiquitination was dramatically increased when

MIR200CHG was silenced (Fig. 7j). Collectively, these findings indicate that MIR200CHG impaired the extending pairing between miR-200c and TDMD-inducing target mRNAs such as ZEB1 by directly binding to the non-seed region of miR-200c, preventing AGO2 from proteasomal degradation to stabilize miR-200c (Fig. 7k).

### MIR200CHG regulates EMT in a partially miR-200c-dependent manner

To examine whether MIR200CHG regulates EMT in a miR-200c-dependent manner, siRNA targeting miR-200c was transfected in MIR200CHG-overexpressed Hs746T cells (Supplementary Fig. 10a). RT-qPCR and western blotting showed that the expression of fibronectin, vimentin, and ZEB1 were decreased, while that of E-cadherin was increased, by MIR200CHG overexpression, which was rescued by the knockdown of miR-200c or by the mutation in miR-200c binding site of MIR200CHG (Supplementary Fig. 10b, c). Wound healing assays revealed that miR-200c inhibition partially abrogated, while MIR200CHG mutant significantly abrogated the defects in cell migration caused by MIR200CHG overexpression in Hs746T cells (Supplementary Fig. 10d). In addition, a miR-200c mimic was transfected when MIR200CHG was knocked down in NCI-N87 cells (Supplementary Fig. 10e). The expression of fibronectin, vimentin, and ZEB1 were increased, while that of E-cadherin was decreased, by MIR200CHG knockdown, which was rescued when miR-200c was overexpressed (Supplementary Fig. 10f, g). Furthermore, increased cell migration caused by MIR200CHG knockdown was effectively alleviated by miR-200c overexpression (Supplementary Fig. 10h). Overall, these findings suggest that miR-200c is a functional downstream target of MIR200CHG and that MIR200CHG regulates EMT in a partially miR-200c-dependent manner.

### Low expression of MIR200CHG is correlated with EMT features and disease progression in patients with GC

To further confirm the correlation between MIR200CHG expression and EMT, RNA FISH of MIR200CHG and immunohistochemistry of E-cadherin and vimentin were performed in 75 paraffin-embedded archived GC tissue samples. The results showed that MIR200CHG was mainly localized in the cytoplasm of tumor cells (Fig. 8a). In addition, MIR200CHG expression was significantly higher in samples with higher E-cadherin expression and lower vimentin expression (Fig. 8a, b). Statistical analysis confirmed a strong association between low MIR200CHG expression and lymph node metastasis (Fig. 8c; Supplementary Data 2). We also observed that patients with advanced clinical stages tended to have lower expression of MIR200CHG (Supplementary Data 2). Collectively, these findings suggest that MIR200CHG may serve as a biomarker for predicting the disease progression of gastric cancer.

### Pan-cancer multi-omics analysis highlighted the functional roles and clinical values of MIR200CHG in multiple cancer types

Finally, we investigated MIR200CHG in all 33 cancers in the TCGA database. Remarkably, co-expression analysis revealed a significant inverse correlation between the promoter methylation and expression of MIR200CHG in 25 cancers (Supplementary Fig. 11a). Besides, we observed a significant positive correlation between the expression of MIR200CHG and miR-200c in 29 cancers (Supplementary Fig. 11b). In contrast, a significant inverse correlation was found between the expression of MIR200CHG and ZEB1, a core EMT marker, consistently across 20 cancers (Supplementary Fig. 11c), suggesting that the function of MIR200CHG in inhibiting the EMT pathway may represent a general mechanism to suppress cancer metastasis. Moreover, the univariate Cox regression analysis revealed MIR200CHG as a prognostic indicator in nine cancers, highlighting its broad translational potential (Supplementary Fig. 11d), which warrants independent validations in the future.

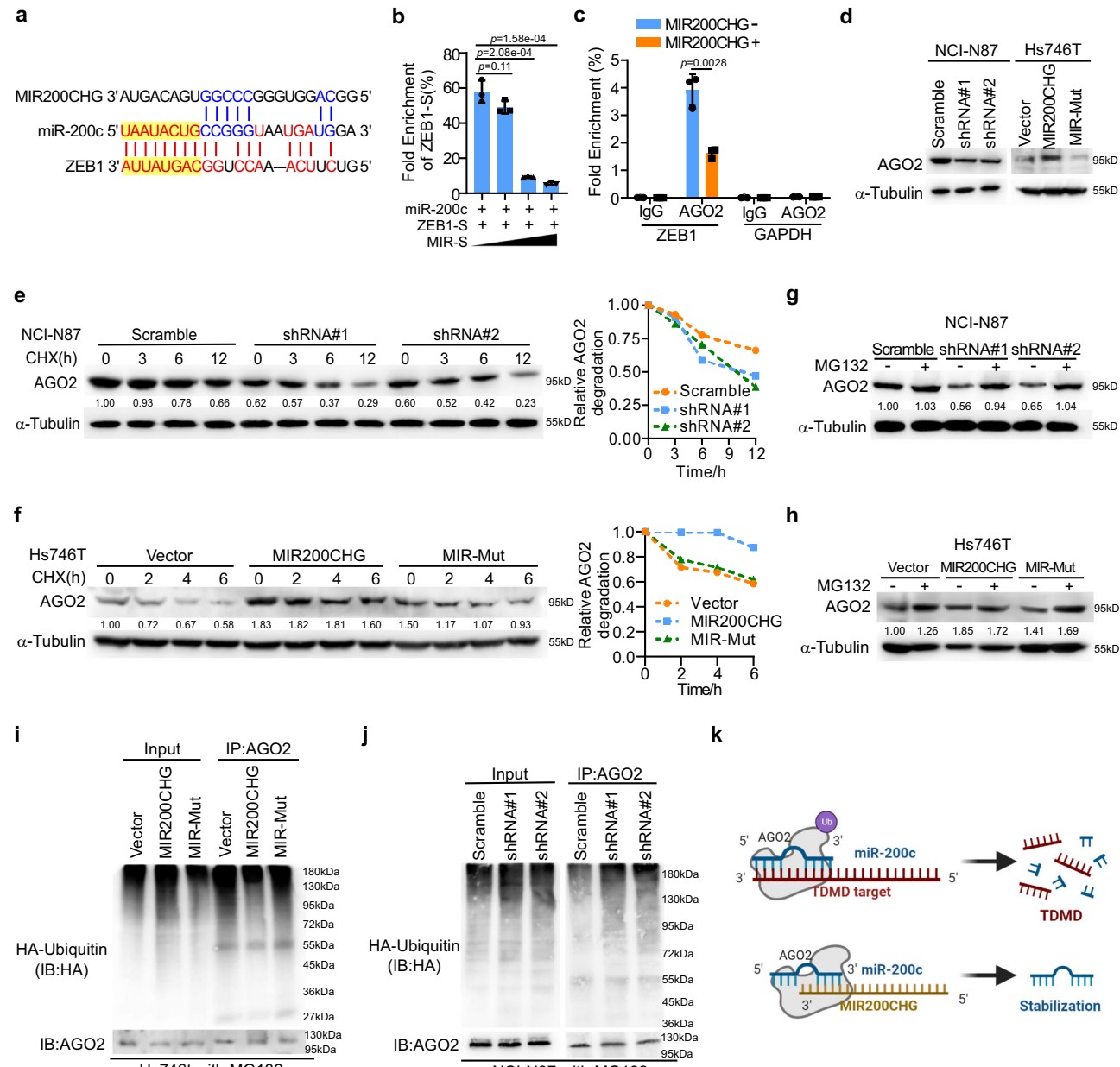

**Fig. 7 | MIR200CHG stabilizes AGO2 from proteasome degradation via competitively binding to the target of miR-200c. a** The sequence match between miR-200c and MIR200CHG or between miR-200c and ZEB1. The seed sequence was highlighted. **b** Gradient doses of MIR200CHG oligos containing the miR-200c binding site (MIR-S) were incubated with miR-200c mimics, and the biotin-labeled ZEB1 mRNA probes (ZEB1-S) were used for RNA-RNA pulldown. Subsequent RT-qPCR analysis showed the interaction of miR-200c with ZEB1. **c** MIR200CHG overexpression plasmid was transfected into Hs746T cells. AGO2-RIP assay and RT-qPCR showed that less ZEB1 occupied the same AGO2 protein when MIR200CHG was present. **d** Western blotting was used to measure the expression of AGO2 in NCI-N87 and Hs746t. **e** Western blotting was used to measure the expression of AGO2 in NCI-N87 following treatment of 20 ug/ml with CHX after the knockdown of MIR200CHG. **f** Western blotting was used to measure the expression of AGO2 in Hs746t following treatment of 20ug/ml with CHX after overexpression of wild-type (MIR200CHG) or mutated MIR200CHG (MIR-Mut). **g**, **h** Western blotting was used

to measure the expression of AGO2 in NCI-N87 and Hs746t that were treated with MG132 after the knockdown or overexpression of wild-type or mutated MIR200CHG. Cells were treated with MG132 (20 μM) for 6 h. **i** Western blot analysis of ubiquitinated AGO2 immunoprecipitated from Hs746t cells with or without wild-type MIR200CHG or MIR200CHG mutation overexpression. The cells were treated with MG132 to inhibit the proteasome. **j** Western blot analysis of ubiquitinated AGO2 immunoprecipitated from NCI-N87 cells with or without MIR200CHG knockdown. The cells were treated with MG132 to inhibit the proteasome. **k** The schematic illustration of the mechanism by which MIR200CHG stabilized miR-200c by inhibiting TDMD (Created with BioRender.com). The experiments were repeated three times independently with similar results (**d**–**j**). Each bar in bar plots represents the mean ± standard deviation of three biologically independent samples (**b**, **c**). *P*-values were determined by two-sided Student's *t*-tests (**b**, **c**). Source data are provided as a Source Data file.

## Discussion

In the last decade, substantial efforts have been made to elucidate the versatile regulatory roles of lncRNAs in various cancers[34–36]. However, most of the existing studies about GC were concerned with individual lncRNAs, lacking a systematic investigation on a genome-wide scale.

Moreover, little is known about the lncRNA transcriptome diversity and subtype-specific regulatory mechanisms. In this study, we first interrogated the lncRNA heterogeneity in GC and found that most of the differential lncRNAs were specific to the MSS/EMT subtype. Using a well-established network-based approach[37–39], we identified lncRNA

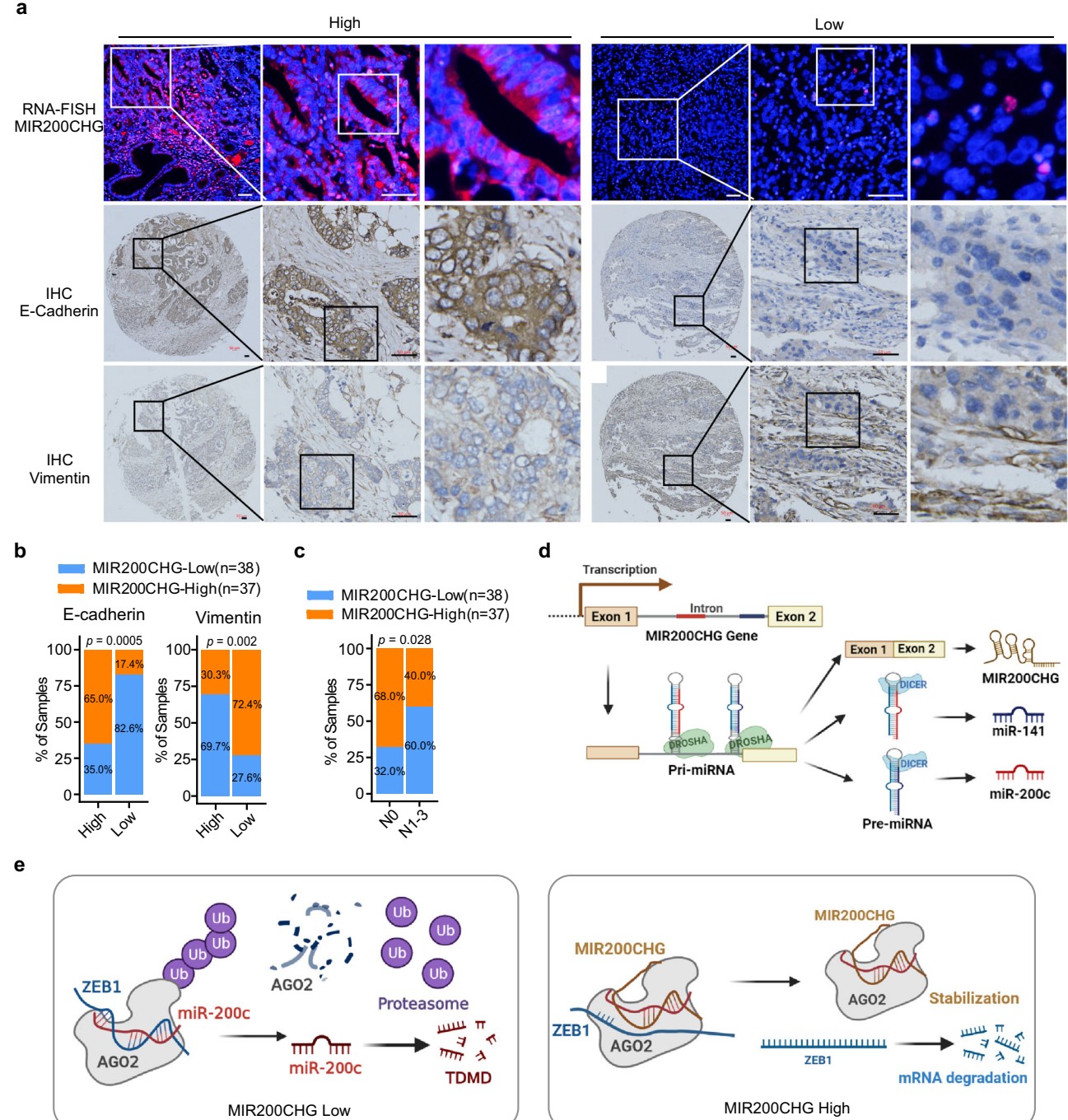

**Fig. 8 | Expression of MIR200CHG and its clinical associations in GC samples.** **a** Representative RNA FISH images of MIR200CHG expression and representative immunohistochemistry images of E-cadherin and vimentin in the same GC tissue microarray (*n* = 75). **b** The association between MIR200CHG and E-cadherin expression (left) and vimentin expression (right) in GC patients (*n* = 75). **c** The associations of MIR200CHG expression with N stage in GC patients (*n* = 75). **d** The schematic illustration shows that MIR200CHG, miR-200c, and miR-141 are derived from the same primary transcript (Created with BioRender.com). **e** The schematic illustration of the mechanism by which MIR200CHG protects miR-200c from TDMD (Created with BioRender.com). *P*-values were based on two-sided Fisher's exact tests (**b**, **c**). Source data are provided as a Source Data file.

MIR200CHG as a master regulator of EMT specifically in the MSS/EMT subtype of GC.

*MIR200CHG* is the host gene of two intronic miRNAs – miR-200c and miR-141 – members of the miR-200 family, which has been demonstrated to inhibit the EMT pathway in cancer cells by directly repressing the EMT-inducing transcriptional factors ZEB1 and ZEB2[40]. For canonical miRNAs, it has been reported that the microprocessing of pri-miRNAs was predominantly co-transcriptional and intronic miRNAs were microprocessed concurrently with intron splicing[29]. In contrast, the microprocessing of intronic miRNAs could facilitate the completion of host gene splicing[41]. Hence, the transcription of *MIR200CHG* facilitates the biogenesis of miR-200c and miR-141 at the transcriptional level (Fig. 8d).

On the other hand, host genes may interact with its intronic miRNAs at the post-transcriptional level, as reported previously[42]. In contrast to the most frequently reported role of cytoplasmic lncRNAs as competing endogenous RNAs, little is known about their roles in miRNA decay. Recently, the TDMD mechanism has been proposed in

miRNA degradation[18]. In addition to the tailing and trimming of miR-NAs during TDMD, ZSWIM8 cullin-RING ubiquitin ligase was recently reported as another independent factor that facilitates TDMD by ubiquitinating AGO2 for degradation by proteasomes and thus exposing the miRNA for degradation[43,44]. In this study, we demonstrated that MIR200CHG directly binds to and stabilizes miR-200c. We provided substantial evidence to show that the direct binding of MIR200CHG in the non-seed region of miR-200c impaired the extended base-pairing of miR-200c and its TDMD-inducing mRNAs such as ZEB1, thus preventing the broad structural rearrangements of AGO2 to stabilizing miR-200c (Fig. 8e). Our findings provide insights into the functional modes of lncRNA in increasing the stability of miRNAs.

Additionally, we confirmed that miR-200c was a functional downstream target of MIR200CHG and that MIR200CHG regulated EMT in a partially miR-200c-dependent manner. Besides, there also exists the possibility that MIR200CHG functions in a miR-200c-independent manner by directly targeting downstream genes, as indicated by the ENCORI database (Supplementary Data 3). miR-200 paralogs, including miR-429, have been implicated in the regulation of EMT in colorectal cancer[45] and hepatocellular carcinoma[46]. Our MS2-RIP results demonstrate that MIR200CHG can interact with miR-429 (Supplementary Fig. 8d), suggesting a potential role for MIR200CHG in EMT regulation through modulation of miR-429 which needs to be further verified.

MIR200CHG (also known as U47924.27) has been investigated in several other malignancies, which seemed to have contradictory findings on its function. Tang et al. reported that MIR200CHG was highly expressed in breast cancer tissues and could promote breast cancer proliferation, invasion, and drug resistance by interacting with and stabilizing YB-1[47], which seems to be contradictory to our conclusion. However, the expression of MIR200CHG was also found to be heterogeneous in breast cancer cell lines in their study. The TNBC cell line MDA-MB-231, which is mesenchymal stem-like and more aggressive, showed little expression of MIR200CHG compared to luminal A cell lines such as MCF7 and T47D. The observation of heterogeneity in the expression of MIR200CHG is highly consistent with our findings in GC. In addition, MIR200CHG has been reported to be a prognostic biomarker by serving as a protective factor[48–50] or risk factor[51] in bladder urothelial carcinoma[48], lung adenocarcinoma[49], melanoma[50], and colorectal cancer[51] based on in silico analysis of public datasets. Therefore, the function of MIR200CHG may be cancer tissue-dependent, cancer subtype-dependent and impacted by other factors such as subcellular localization. Furthermore, the mesenchymal subtype exists in many cancers and was recently found to be heterogeneous in cancers such as glioblastoma[52]. Although MIR200CHG was differentially expressed between MSS/EMT and other subtypes of GC, heterogeneity does exist within the MSS/EMT GC subtype (Supplementary Fig. 12). Therefore, the biological functions and regulatory mechanisms of MIR200CHG rely on more in-depth investigations in a more specific context.

In conclusion, we identified MIR200CHG as a master regulator of EMT in the MSS/EMT subtype of GC. The repressed expression of MIR200CHG in the MSS/EMT subtype was attributed to the hypermethylation of its promoter, and the deficiency of MIR200CHG induced the mesenchymal phenotype of GC cells. Mechanistically, the transcription of *MIR200CHG* facilitates the biogenesis of its intronic miRNAs – miR-200c and miR-141 – at the transcriptional level, and stabilizes miR-200c at the post-transcriptional level by directly binding to the non-seed region of miR-200c to prevent its TDMD, thus inhibiting the EMT process. Our findings provide solid evidence supporting MIR200CHG as a potential prognostic and predictive biomarker as well as a therapeutic target for GC.

## Methods
### Ethics statement
This research complies with all relevant ethical regulations.

### Public dataset curation and pre-processing
Genome-wide mRNA and lncRNA expression profiles, 450 K DNA methylation microarray data, miRNA expression data, and corresponding clinical information from the 'STAD' project of the Cancer Genome Atlas (TCGA-STAD) were downloaded using the R package 'TCGAbiolinks'[53]. Gene expression data, measured as Fragments Per Kilobase of transcript per Million mapped reads (FPKM), was filtered by removing genes without any expression in any samples, deduplicated by keeping the gene with the maximum mean expression value across all samples, and log2 transformed. Next, we downloaded the latest lncRNA annotation from the GENCODE project[54] and extracted the lncRNA expression from the TCGA-STAD dataset. GC molecular subtyping labels were obtained from ref. 3. Finally, 179 samples with corresponding mRNA expression, lncRNA expression, DNA methylation, miRNA expression, and subtype labels were retained for further analysis. In addition, we downloaded two other datasets for validation: GSE62254 (also known as the ACRG cohort; $n = 300$) and GSE15459 ($n = 192$) from Gene Expression Omnibus (GEO) using the R package 'GEOquery'[55]. The corresponding survival information and ACRG subtyping labels for these two datasets were also obtained from Cristescu et al.[3].

### Differential gene expression analysis and gene set enrichment analysis (GSEA)
Differential mRNA expression analysis between each subtype of GC and the others was performed using the R package 'limma'[56], respectively, for each of the three datasets (TCGA, ACRG, and GSE15459). Meanwhile, differential lncRNA expression analysis was performed between the MSS/EMT subtype and non-MSS/EMT subtypes in the TCGA cohort also using 'limma' for further network analysis. GSEA was performed using the R package 'HTSanalyzeR2' with permutation tests[57]. Pre-defined gene sets, including curated cancer-related signatures, canonical pathways, metabolic pathways, and immune cell signatures, were used for the analysis[58]. P-values indicating the significance of enrichment were estimated by 10,000 permutations. Significant gene sets were selected based on Benjamini−Hochberg (BH)-adjusted p-values lower than 0.05. The heatmap of GSEA result was generated using the R package 'pheatmap'.

### LncRNA−mRNA regulatory network inference and master regulatory analysis
To perform regulatory network inference, top differentially expressed (DE) lncRNAs ($|\log2FC| > 1$, BH-adjusted $P < 1 \times 10^{-5}$) and mRNAs ($|\log2FC| > 0.5$, BH-adjusted $P < 1 \times 10^{-5}$) between the MSS/EMT subtype and non-MSS/EMT subtypes were first selected. The expression profiles of top DE lncRNAs and mRNAs were merged for regulatory network inference using the R package 'RTN' (v2.12.1)[59]. The final regulatory network was visualized using the R package 'RedeR'[60]. Master regulator analysis was performed based on hypergeometric tests for overrepresentation of GC EMT signature genes[3] in the regulon of each lncRNA using the R package 'RTN' (v2.12.1)[59]. P-values were corrected for multiple hypothesis testing using the BH procedure.

### Cell line expression data curation and pre-processing
The gene expression profiles of GC cell lines were downloaded from the Cancer Cell Line Encyclopedia (CCLE) database[26]. Gene expression data measured as Transcripts Per kilobase Million (TPM) were log2 transformed. Molecular subtyping labels for the GC cell lines from the CCLE database were obtained from ref. 3. The expression profiles for six MSS/EMT cell lines and 31 non-MSS/EMT cell lines were retained for further analysis.

### MIR200CHG promoter methylation analysis
Due to different genome builds used by the methylation data (hg19) and lncRNA annotations in the TCGA cohort, the genome coordinates

were first unified using the R package 'liftOver'. The promoter DNA methylation of MIR200CHG was calculated by the mean of all annotated CpG sites within the transcription start site ±1 kb. Pearson's correlation coefficients (PCCs) were calculated to quantify the associations between promoter DNA methylation levels and MIR200CHG expression levels. Linear regression models were fitted for visualization. Similar analyses were performed for GC cell lines in the CCLE database.

### Drug sensitivity analysis
Drug sensitivity data for GC cell lines were downloaded from the CCLE database. Pearson's correlation analysis was performed between MIR200CHG expression levels and half-maximal inhibitory concentration (IC50) values. Drugs with $P < 0.05$ were selected for further analysis.

### Pan-cancer analysis of MIR200CHG
The gene expression profiles, miRNA expression profiles and DNA methylation profiles for 33 cancers in the TCGA database were downloaded using the R package 'TCGAbiolinks'[53]. The clinical data containing survival information for 33 cancers were downloaded from ref. 61. Both the gene and miRNA expression profiles for each cancer were log2 transformed. PCCs were calculated to quantify the associations between promoter DNA methylation levels and MIR200CHG expression levels in each cancer. Pearson's correlation analysis was performed between MIR200CHG expression and miR-200c expression or ZEB1 expression in each cancer. Univariate Cox regression analysis was performed for each TCGA cancer using the R package 'survival'.

### Cell culture
The GC cell lines SNU-668, MKN1, NUGC-4, NCI-N87, SNU-620, Hs746T, OCUM-1, and SNU-719 were purchased from ATCC (Manassas, VA, USA). The GC cell lines SNU-668, MKN1, NUGC-4, NCI-N87, and SNU-620 were cultured in RPMI1640 medium supplemented with 10% fetal bovine serum (FBS). Other cell lines – Hs746T, OCUM-1, and SNU-719–were cultured in complete Dulbecco's modified Eagle medium supplemented with 10% FBS. The human embryonic kidney cells (HEK293FT) were obtained from the Cell Bank of Shanghai Institutes of Biological Sciences (Shanghai, China) and cultured in Dulbecco's modified Eagle's medium (DMEM) with 10% FBS. All of the cell lines were incubated at 37 °C in a 5% $CO_2$ atmosphere and were grown to 50–80% confluence before the next passage or further experiments. All cell lines used in this study were tested to confirm that they were free of mycoplasma contamination.

### Lentivirus production and transduction
To construct a recombinant lentiviral vector expressing MIR200CHG, the full-length gene of MIR200CHG was inserted downstream of the Ubi promoter in the Ubi-MCS-EGFP-IRES-Puromycin vector (Genechem, Shanghai, China). An empty pUbi-MCS vector was used as control. A short hairpin RNA (shRNA) targeting MIR200CHG was synthesized and subcloned into the hU6-MCS-Ubiquitin-EGFP-IRES-puromycin vector (Genechem, Shanghai, China). shRNA targeting sequences were shown in Supplementary Table 3. HEK293T cells were transfected with the vector and the lentiviral vector packaging system to produce lentiviruses. A scramble shRNA served as the control. Hs746T, SNU668, NCI-N87, and NUGC4 cells were infected with the corresponding concentrated lentiviruses and were selected with puromycin to establish cell lines with stable overexpression or knockdown of MIR200CHG, which was verified by RT-qPCR.

### RNA extraction, reverse transcription, and quantitative PCR
Cultured cells were harvested in TRIzol reagent (Invitrogen, Carlsbad, CA, USA) for RNA extraction. One microgram of RNA was used for cDNA synthesis primed with random hexamers. RT-qPCR was performed using the ChamQ Universal SYBR qPCR Master Mix (Vazyme, Nanjing, China) on the Bio-Rad CFX96 Real-Time System C1000 Cycler (Bio-Rad Laboratories, Singapore). Data were analyzed with CFX ManagerV2 software using GAPDH expression data for normalization. Each experiment was performed in triplicate. The primers used for RT-qPCR were listed in Supplementary Table 4.

### Stem-loop RT-qPCR for miRNAs
Total RNA was isolated from gastric cancer cell lines using Trizol reagent (Invitrogen, USA) following the manufacturer's protocol. Target-specific cDNA synthesis proceeds with the stem-loop RT primer and then qPCR subsequently proceeds with the forward primer to rapidly achieve a quantitative result. We synthesized cDNA by reverse transcription reaction using a miRNA 1st Strand cDNA Synthesis Kit (by stem-loop) (Vazyme #MR101-01, China). qPCR was conducted using a miRNA Universal SYBR qPCR Master Mix (Vazyme #MQ101, China) according to manufacturer's instructions. Results were normalized using U6 as an internal control. To account for the assessment of technical variability, the assays were performed in triplicate for each case. Stem-loop RT primers and qPCR primers are designed by Vazyme miRNA Design V1.01. and sequences are shown in Supplementary Table 4.

### Cytoplasmic and nuclear fractionation
Subcellular fractionation of cells was performed as described previously[62]. Cytoplasmic and nuclear RNAs of NCI-N87 cells were isolated and purified using the Cytoplasmic and Nuclear RNA Purification Kit (BestBio, BB-3104).

### RNA fluorescence in situ hybridisation (FISH)
Cy3-labeled MIR200CHG probes were obtained from GenePharma (Shanghai, China). The probe sequences were listed in Supplementary Table 3. NCI-N87 cells grown on slides were washed with phosphate-buffered saline (PBS) and fixed in 4% paraformaldehyde. Subcellular localization of MIR200CHG was detected using the cellular RNA FISH Kit (GenePharma, Shanghai, China). MIR200CHG expression in paraffin-embedded GC tissues was detected using the tissue RNA FISH Kit (GenePharma, Shanghai, China). ImageJ software was used for Cy3 fluorescence quantification. The samples were divided into two groups with high and low MIR200CHG expression according to the median value of Cy3 fluorescence intensity.

### Western blotting
Cells at 70–80% confluence were washed twice with ice-cold PBS and harvested in the sample buffer supplemented with the Protease/Phosphatase Inhibitor Cocktail (Cell Signaling Technology). Protein levels were determined using the bicinchoninic acid assay (Pierce, Rockford, IL, USA) according to the manufacturer's instructions. Equal amounts (10 μg) of protein were separated using sodium dodecyl sulfate-polyacrylamide gel electrophoresis and transferred to polyvinylidene difluoride membranes (Bio-Rad). After blocking in 5% milk in Tris-buffered saline with 0.1% Tween-20, the membranes were incubated with primary antibodies followed by horseradish peroxidase-conjugated secondary antibodies. The following antibodies were used: anti-E-cadherin rabbit polyclonal antibody (1:5000, Proteintech, 20874-1-AP), anti-vimentin rabbit polyclonal antibody (1:5000, Proteintech, 10366-1-AP), anti-ZO-1 mouse monoclonal antibody (1:1000, Proteintech, 66452-1-Ig), anti-fibronectin monoclonal antibody (1:1000, Bioworld, BS1644), and anti-α-tubulin mouse monoclonal antibody (1:10,000, RayAntibody, RM2007L). The antibodies used for western blot were listed in Supplementary Table 5.

## Transwell migration assay and wound healing assay

A Boyden chamber with an 8-μm-pore filter membrane was used for the in vitro migration and invasion assay. Briefly, cells ($1 \times 10^5$) in the culture medium without FBS were seeded in the upper chamber, and the culture medium with 20% FBS was added to the lower chamber as a chemoattractant. After incubation for 24 h, cells on the upper side of the filter were removed with cotton swabs. Cells that migrated to the lower surface of the filter were fixed in 4% paraformaldehyde and stained with crystal violet. The migratory cells were counted (five random fields per well under 200× magnification) by an inverted microscope. Three independent experiments were performed, and the data were presented as the mean ± standard deviation. For the wound healing assay, wounds were scratched in confluent cells using a pipette tip, and the cells were then rinsed with PBS. Serum-free medium was then added, and culture plates were incubated in an IncuCyte ZOOM™ incubator. Cell cultures were imaged at 0 and 72 h and the wound confluence was calculated by IncuCyte system.

## Immunofluorescence

For immunofluorescence analysis, cells were seeded on coverslips, cultured for 48 h, fixed with 4% formaldehyde for 10 min at room temperature (RT), and washed thrice with wash buffer (0.02% Tween-20/PBS). The cells were permeabilized with 0.5% Triton X-100/PBS for 5–10 min at RT, then washed with the wash buffer three times (5 min each time), and then incubated with 1.5% bovine serum albumin/PBS solution (blocking solution) for 30 min at RT. Subsequently, the cells were incubated with anti-E-cadherin (1:200, Proteintech, 20874-1-AP), anti-ZO-1 (1:500, Proteintech, 66452-1-Ig), anti-fibronectin (1:200, Bioworld, BS1644), and anti-vimentin (1:500, Proteintech, 10366-1-AP) antibodies in blocking solution at 4 °C overnight. Rhodamine phalloidin (1:5000, Solarbio, CA1610) used for detecting F-actin, was added to the cell in a blocking solution, and followed by further incubation in the dark for 60 min at RT. After washing, the cells were incubated with DAPI (Thermo Fisher) in the dark for 15 min at RT. Samples were then mounted with ProLong Gold antifade reagent (LifeTechnologies) and imaged using a fluorescence microscope. Fluorescence intensity was analyzed by ImageJ (version 1.37).

## Mouse lymphatic metastasis model and peritoneal metastasis model

Female BALB/c nude mice and NSG mice (4–5 weeks old, 18–20 g) were purchased from the Experimental Animal Center, Southern Medical University (Guangzhou, China), and housed in a barrier facility on a 12 h light/dark cycle at 20–22 °C and 40–50% humidity. All animal experiments were conducted after receiving approval from the Ethnics Committee of Southern Medical University (No. L2019023). For lymphatic metastasis model, cells ($5 \times 10^6$) were injected into the footpads of the nude mice. On day 60, the mice were euthanized, and primary tumors and inguinal lymph nodes (LNs) were collected for immunohistochemistry. For peritoneal metastasis model, $5 \times 10^6$ cells were injected to abdominal cavity of NSG mice under general anesthesia. Ten mice from each group were measured twice a week and their survival was observed until the mice died. Forty days post-injection, six mice from each group were sacrificed and peritoneal tissue samples were collected for further analysis. The ethics committee specified that the maximal tumor burden is no more than 10% of the body weight of animals and the average diameter is less than 20 mm. During the experiment, the tumor sizes of the mice complied with the regulations.

## Patient information and tissue specimens

We collected a tissue microarray containing 75 gastric cancer tissues from the Biobank of Shanghai OUTDO Biotechnology Co., LTD. The clinicopathological characteristics, including details of the covariate-related population characteristics of human research participants (such as age, gender, etc.), are summarized in Supplementary Data 2.

Patients provided written informed consent, and the study was carried out according to the Declaration of Helsinki. Ethics approval (#YB M-05-01) was obtained from the Institutional Research Ethics Committee of Shanghai Outdo Biotech Company to use the clinical specimens for research purposes.

## Immunohistochemistry (IHC)

IHC for panCK (ABcam, ab7753), E-cadherin (Proteintech, 20874-1-AP), and vimentin (Proteintech, 10366-1-AP) was performed as previously described[63]. The degree of E-cadherin and vimentin IHC staining was reviewed and scored based on the proportion of positively stained stromal cells. The degree of IHC staining was reviewed and scored independently by two observers based on both the proportion of positively stained tumor cells and the intensity of staining. The proportion of tumor cells was scored as follows: 0 (no positive tumor cells), 1 (<10% positive tumor cells), 2 (10–50% positive tumor cells), and 3 (>50% positive tumor cells). The intensity of staining was graded as follows: 0 (no staining); 1 (weak staining = light yellow), 2 (moderate staining = yellow-brown), and 3 (strong staining = brown). The staining index was calculated as the staining intensity score × the proportion of positive tumor cells. Using this method of assessment, we evaluated the expression of E-cadherin and vimentin by determining the staining index based on scores of 0, 1, 2, 3, 4, 6, and 9. Optimal cut-off values were identified: a staining index ≥ 4 was used to define tumors with high expression, and an index ≤ 3 was used to define tumors with low expression.

## MS2 RNA immunoprecipitation (RIP) assays

pcDNA3.1-MS2-12X, pcDNA3.1-MS2-MIR200CHG-WT or pcDNA3.1-MS2-MIR200CHG-Mut was co-transfected with pMS2-GFP (Addgene) into HEK293T cells for 48 h. RIP assays were performed as described previously[64] using the GFP antibody (Proteintech, 66002-1-Ig). The immunoprecipitated RNA was subjected to RT-qPCR detection.

## AGO2 RIP assays

HEK293T cells were transfected with miR-200c mimics or MIR200CHG for 48 h. Transfection cells were washed in cold PBS, scraped in PBS, and collected by centrifugation. Pellets were then resuspended in lysis buffer (20 mM Tris-HCl pH 7.5, 150 mM KCl, 0.5% NP40, 2 mM EDTA, 1 mM NaF, 0.5 mM DTT, and 160 U/mL RNasin, protease, and phosphate inhibitors) and precleared with Protein G sepharose beads (Invitrogen) for 1 h at 4 °C. Part of the cleared lysates (10%) was used as input, and the remainder was incubated with Protein G sepharose beads conjugated with anti-AGO2 (Proteintech,67934-1-Ig) or IgG (DIA-AN, Q6004) antibodies for 4 h at 4 °C. After washing, 10 μL of the immunoprecipitate was kept for western blot analysis, and the remainder was treated with DNase I and proteinase K for 20 min at 50 °C. RNA was extracted using phenol-chloroform and ethanol–sodium acetate precipitation and then quantified using Nanodrop. The antibodies used in this study were listed in Supplementary Table 5.

## In vitro translation

Biotin-labeled RNA probes and transcripts used in the RNA pulldown and RNA-RNA pulldown assays were generated by in vitro transcription. Transcription assays were performed using T7 High Yield Transcription Kit (Vazyme, Nanjing, China) according to the manufacturer's instructions. Biotinylated nucleotide was attached to the 3' terminus of RNA strand using Pierce™ RNA 3' End Biotinylation Kit (Thermo Scientific).

## RNA pull-down

The RNA oligonucleotides with 3' biotin modifications were synthesized by Integrated DNA Technologies. The RNA sequences were shown in Supplementary Table 3. Biotinylated RNA oligos

(400 pmol) were immobilized to streptavidin beads before incubation with HEK293FT cell extracts in binding buffer (20 mM Tris, 200 mM NaCl, 6 mM EDTA, 5 mM potassium fluoride, 5 mM b-glycerophosphate, 2 mg/mL aprotinin at pH 7.5) overnight at 4 °C. Beads were washed three times with binding buffer and boiled in sodium dodecyl sulfate buffer. The samples were analyzed by Western blot.

### RNA-RNA interaction assay
For in vitro pulldown assay, full-length MIR200CHG, MIR200CHG-S, and ZEB1-S RNA were transcribed and purified in vitro. An equal amount of RNA transcripts was used for in vitro binding assay. Denatured RNA transcripts were incubated with prewashed streptavidin agarose beads in an RNA interaction buffer (50 mM sodium cacodylate, pH 7.5; 300 mM KCl; and 10 mM MgCl2) for 37 °C for 1 h. The binding complexes were pulled down by biotin-labeled RNA probes. The pulled-down RNAs were quantified by RT-qPCR.

### Immunoprecipitation and ubiquitination analysis
Cells were transfected with HA-ubiquitin, then were treated with MG132 (20 μM) for 12 h. The cells were collected and washed three times with pre-cooled PBS, lysed with lysis/washing buffer (0.025 M Tris, 0.15 M NaCl, 0.001 M EDTA, 1% NP40, 5% glycerol), and 10 μL of the cell lysate was taken as the input. Cell lysates were incubated with magnetic beads conjugated with negative control normal mouse IgG or human anti-Ago2 antibody overnight at 4 °C with rotation. Subsequently, the tube was put on a magnetic stand and the supernatant was discarded, followed by washing the magnetic beads thoroughly with the lysis/washing buffer. Then, 40 μL of 1× SDS-PAGE loading buffer was added, boiled for 5 min, and finally the proteins were detected by western blot.

### Statistical analysis
All results are presented as the mean ± standard error of the mean. Wilcoxon rank-sum tests and one-way analyses of variance (ANOVA) were used to evaluate the significance of the differences between different groups. Univariate and multivariate Cox proportional hazard regression analysis and survival analysis were performed using the R package 'survival'. Kaplan–Meier plots with log-rank tests were performed to compare patient groups stratified by the median value of the lncRNA expression level. $P$-values lower than 0.05 were considered statistically significant.

### Reporting summary
Further information on research design is available in the Nature Portfolio Reporting Summary linked to this article.

## Data availability
The genome-wide mRNA and lncRNA expression profiles, 450 K DNA methylation microarray data, miRNA expression data, and clinical information of 'TCGA-STAD' are available in The Cancer Genome Atlas (TCGA, https://portal.gdc.cancer.gov/). The latest lncRNA annotation for human is available from the GENCODE project (https://www.gencodegenes.org/human/). The gene expression profiles, miRNA expression profiles and DNA methylation profiles for 33 cancers are available in the TCGA database. Two independent gastric cancer gene expression datasets are available in Gene Expression Omnibus (GEO, https://www.ncbi.nlm.nih.gov/geo/) database under the accession code GSE62254 and GSE15459. The gene expression profiles and drug sensitivity data of gastric cancer cell lines are available in the Cancer Cell Line Encyclopedia (CCLE, https://sites.broadinstitute.org/ccle/datasets). The remaining data are available within the article and supplementary information. Source data are available as a Source Data file. Source data are provided with this paper.

## Code availability
All codes used in this study are available at https://github.com/CityUHK-CompBio/GC-MIR200CHG and have also been deposited on Zenodo: https://zenodo.org/records/10051102[65].

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

## Acknowledgements

This work was funded by grants from the National Natural Science Foundation of China (No. 82173289 (W.L.), 81872401 (W.L.), 82173058 (H.H.)), grants from Guangdong Basic and Applied Basic Research Foundation (Project No.2019B030302012 (X.W.), 2021A1515010425 (H.H.)), a grant from Shenzhen Science, Technology and Innovation Commission (Project No. 基 2020N368, X.W.), a startup grant (Project No. 4937084, X.W.), direct grant (2021.077, X.W.), Faculty Postdoctoral Fellowship Scheme 2021/22 (Project No. FPFS/2122/32, X.W.) from the Chinese University of Hong Kong, and grants from the Research Grants Council (Project No. C4024-22GF, 14104223, 11103619, 11103921, 14111522, R4017-18) of the Hong Kong Special Administrative Region, China, awarded to X.W. This work was also partially sponsored by Shenzhen Bay Scholars Program awarded to X.W.

## Author contributions

X.W., Y. Zhu, W.L., and C.H. conceived the project and designed the research. Y. Zhu performed the bioinformatics analysis. C.H., C.Z., Y.Z., Y. Zhang, X.P., and E.Z. performed the experiments. Y. Zhu and C.H. generated the figures and tables, and drafted the manuscript. X.W., Y. Zhu, W.L., and C.H. interpreted the data and discussed the results. X.W., W.L., and H.H. provided funding and material support, study supervision and manuscript revision. All authors read and approved the final manuscript.

## Competing interests

The authors declare no competing interests.
