## [Peer Review File · Nature Communications]

LncRNA MIR200CHG inhibits EMT in gastric cancer by stabilizing miR-200c from target-directed miRNA degradationREVIEWER COMMENTS

Reviewer #1 (Remarks to the Author):

This manuscript investigates whether lncRNAs (long non-coding RNAs) might be contributing to EMT (epithelial-mesenchymal transition) in gastric cancer. A computational analysis implicates the lncRNA MIR200CHG as a potential regulator of EMT in gastric cancer. This lncRNA harbors the microRNA miR-200 (and miR-141), and is also proposed to protect miR-200 from degradation. A series of assays offer some support for the computational inferences. Data suggests a complex series of interactions between the miRNAs, the lncRNAs, targets and AGO (trans factors that enable miRNA functions). Some aspects of the model are more convincing than others (See below). Perhaps the key concern is the degree to which the functional data rests upon roles for the lncRNA or the miRNAs within? Adding additional data on this point is most important. Overall, this is an interesting paper, which would likely be significantly improved if all (or most) of the points below were addressed, after which it would certainly warrant publication in Nat. Communications.

Major

1. Additional data/analysis is needed to corroborate that the inferred network is 'robust'. For example, is the regulon supported by the ACRG and GSE15459 datasets (after defining the regulon using the TCGA dataset alone).
2. Similarly, what fraction of the MIR200CHG regulon overlaps with predicted targets of miR-200 (or miR-141)
3. Section beginning line 289 – the conclusions made would be far more robust if MIR200CHG overexpressed cells were generated that contained mutations that disrupt miR-200 and/or miR141. This is an important experiment that would allow the relative contributions of the lncRNA and the miRNA be differentiated (which is a large part of the novelty and purpose of the paper).
4. Code used in this study should be made available with the paper (e.g., through Github). This is clearly preferable to 'available upon request'
5. Data in Figure 3E-G is not particularly convincing – I agree that the correlations are apparent, but the text overstates the potential significance of methylation. Suggest a more balanced description of these data (or adding additional functional experiments).
6. Repeating some of the experiments in Figures 4 and 5 with a MIR200CHG variant that disrupts each (and both) of the two miRNAs would greatly strengthen the conclusions made in the paper.
7. The apparent pairing between the lncRNA and miR-200 is rather minimal and less than what would be expected given the model – this could be directly tested by mutating the binding site.

Minor

1. There are publications linking lncRNAs to miRNAs via TDMD – these should be cited in the introduction.
2. The authors should explicitly state that the initial analyses use transcriptomic data (as they introduce the approach).
3. Line 160, remove 'strongly'
4. Line 219, change 'synergic/cooperative' to 'as a consequence of MIR200CHG transcription' None of the data (in this section) speaks to cooperativity nor synergism.
5. Figure 7A – do the '...' in ZEB1 indicate missing nucleotides, if so, how many?
6. Are other miR-200 paralogs expressed in these cells? If so, how does the model accommodate such expression?

Reviewer #2 (Remarks to the Author):

The association of MIR200CGH has been reported in multiple cancer types (renal cell carcinoma, urothelial cancer, breast cancer). Its role seems to have varied. For example, in bladder cancer, OE is associated with poor prognosis (contradicts your results), similarly, no expression produces poor outcome in breast cancer (similar to your results). Prior publications have stated that MIR200CGH may be an oncoMiR or TSGMiR depending on the context. This issue is serious and needs to be addressed.

Your data shows that MIR200CHG is down regulated (poor prognosis) in gastric cancer. Once an LNC has low expression or loss of expression, it would be difficult to target therapeutically, yet you did not seriously pursue the two that are upregulated in gastric cancer (AC104083.1 and LINC00578). Pursuing these could be interesting and potential worthy of pursuit in the clinic.

You emphasized heterogeneity of gastric cancers but following the MIR200CHG research trajectory, heterogeneity concept is lost. MSS/EMT subtype was chosen but one is sure it is a heterogeneous subtype. Mesenchymal subtype have been described more than ones. It is unclear if there is a subtype associated with OE of MIR200CHG and another one with its low expression. Overall, agree that MIR200CHG can serve as a prognostic marker (how many more do we need in gastric cancer) but if this can not be further pursued then what to become of it.

There is little mention of published literature on MIR200CHG in cancers. and discordant results should be explained by generating a pan-cancer integrated signature that might shed some light on its function in various contexts.

Was MIR200CHG explored in other gastric cancers (other than MSS/EMT), this is not clear.

So ultimately, we have low expression of MIR200CHG. Various functional experiments to document that hypermethylation (which is often the case for TSGs in cancer in general, some exceptions noted), EMT can be reversed if MIR200CHG is induced, the lymph node metastases model is suboptimal, and its binding and stabilization of miR-200c, and its prognostic value. The pan-cancer analysis is sub optimally described and contradictions exist and does not address published literature.

Multivariate analysis is left out.

No new targets and no drug

No human gastric cancer models

Reviewer #3 (Remarks to the Author):

In the current manuscript, the authors investigated the subtype-specific expression and general function of the non-coding RNA MIR200CHG in gastric cancer (GC). This ncRNA is significantly repressed in the MSS/EMT subtype of GC which seems to be due to epigenetic silencing, i.e. promoter hypermethylation. Overexpression of MIR200CHG inhibited migration and invasion of GC cells of the MSS/EMT subtype *in vitro* and metastasis *in vivo* whereas knockdown of the ncRNA in GC cells of other subtypes induced an epithelial-mesenchymal transition and enhanced cell motility. Mechanistically, the authors show that the spliced MIR200CHG is able to bind to the intronically encoded mature miR-200c, but not to miR-141 that is also derived from the same intron. The direct RNA:RNA interaction seems to stabilize miR-200c by preventing it from binding to its downstream mRNA targets. In particular, the interaction between miR-200c and Zeb1 mRNA is reduced upon MIR200CHG overexpression. This in turn prevents the target-dependent miR-200c decay. Overall, the study follows a logical flow and the manuscript is well-written. The authors assembled lots of data and included several important controls in their assays. Therefore, I only have some minor concerns that should be addressed:

- 1) While most assays contain relevant and important controls some do not. For example, all pulldown assays lack a proper specificity control, i.e. a non-binding negative control. For instance, Figure 7C shows the reduced binding of Zeb1 to Ago2 upon MIR200CHG overexpression as expected. The authors should show the enrichment of other mRNAs, either with or without miR-200c binding sites. Furthermore, RNA FISH assays (Suppl. Figure 4A) lack negative controls, i.e. cells that do not express MIR200CHG.
- 2) The RNA FISH analysis (Suppl. Figure 4A) suggests that MIR200CHG localizes in cytoplasmic speckles. Do these speckles contain Ago2? Please show co-localization. Do the authors have any idea what the nature of these speckles might be? Stress granules?
- 3) In Figure 6F, the authors should include an additional control and repeat the assay with miR-141 mimics as well.
- 4) In Figure 6C (and also in other stability assays shown in the Supplementary Data), the half-life of the RNA of interest (here: miR-200c) varies dramatically between both controls (vector control, shScramble). The changes are within the range of the experimental groups. Any explanation for this variation? In theory, the half-life should not be altered by the controls.
- 5) Does the modulation of Zeb1 expression affect miR-200c stability?
- 6) It is unclear how the miR-S probe depicted in Suppl. Figure 6J, which is complementary to the MIR200CHG but not miR-200c, should be able to bind to endogenous miR-200c as stated in line 244ff and shown in Figure 6G and 6H. Please describe this assay in more detail and check the sequence and/or wording.
- 7) Please provide a more detailed description of the miRNA quantification (stem-loop RT-PCR protocol).

Point by point response to reviewers' comments:

We sincerely appreciate the reviewers for the positive comments on our work and the highly valuable suggestions. We have addressed all comments as follows and made revisions accordingly in the updated manuscript.

Reviewer #1 (Remarks to the Author)

This manuscript investigates whether lncRNAs (long non-coding RNAs) might be contributing to EMT (epithelial-mesenchymal transition) in gastric cancer. A computational analysis implicates the lncRNA MIR200CHG as a potential regulator of EMT in gastric cancer. This lncRNA harbors the microRNA miR-200 (and miR-141), and is also proposed to protect miR-200 from degradation. A series of assays offer some support for the computational inferences. Data suggests a complex series of interactions between the miRNAs, the lncRNAs, targets and AGO (trans factors that enable miRNA functions). Some aspects of the model are more convincing than others (See below). Perhaps the key concern is the degree to which the functional data rests upon roles for the lncRNA or the miRNAs within? Adding additional data on this point is most important. Overall, this is an interesting paper, which would likely be significantly improved if all (or most) of the points below were addressed, after which it would certainly warrant publication in Nat. Communications.

Major

1. Additional data/analysis is needed to corroborate that the inferred network is 'robust'. For example, is the regulon supported by the ACRG and GSE15459 datasets (after defining the regulon using the TCGA dataset alone).

Response: We thank the reviewer for the insight. Due to the lack of lncRNA expression profiles, the regulatory network inference could not be performed on the ACRG and GSE15459 (microarray-based) datasets. Instead, we investigated the expression of MIR200CHG regulon (332 genes) in the ACRG and GSE15459 datasets, respectively. As shown in **Figure R1A (new Supplementary Figure 2A)**, the MIR200CHG regulon showed consistent expression pattern across GC subtypes in the TCGA, ACRG and GSE15459 datasets. Moreover, genes induced by MIR200CHG (92 genes) defined in the TCGA dataset were more enriched in the non-MSS/EMT subtype in both the ACRG and GSE15459 datasets (**Figure R1B, D, new Supplementary Figure 2B, D**). On the contrary, genes repressed by MIR200CHG (240 genes) were more enriched in the MSS/EMT subtype in both the ACRG and GSE15459 datasets (**Figure R1C, E, new Supplementary Figure 2C, E**). We've added these results to the revised manuscript (Lines 105-114).

Figure R1. (A) Heatmaps compare the MIR200CHG regulon expression patterns in the TCGA, ACRG and GSE15459 datasets. Rows are sorted by the same order as in TCGA. (B-E) Gene set enrichment plots of MIR200CHG induced and repressed genes in the ACRG and GSE15459 datasets, respectively. The upper panel illustrates the running sum scores of GSEA random walks, the middle and lower panels show the positions of the MIR200CHG induced and repressed genes in the gene list ranked by log₂ fold change between MSS/EMT and non-MSS/EMT in the ACRG (B, C) and GSE15459 (D, E) datasets.

2. Similarly, what fraction of the MIR200CHG regulon overlaps with predicted targets of miR-200 (or miR-141)

Response: We thank the reviewer for the comment. As the MIR200CHG regulon was defined by network inference based on functional associations of gene expression, we performed similar network inference using mRNA and miRNA expression profiles to predict the targets of miR-200c and miR-141. As a result, we found there were 714 and 845 genes in the miR-200c and miR-141 regulons, with significant overlaps with the regulon of MIR200CHG ($P = 1.15e-17$ and $P = 5.38e-16$, hypergeometric tests, **Figure R2**).

Figure R2. Venn diagrams show the overlaps (A) between the regulons of miR-200c and MIR200CHG, and (B) between the regulons of miR-141 and MIR200CHG.

3. Section beginning line 289 – the conclusions made would be far more robust if MIR200CHG overexpressed cells were generated that contained mutations that disrupt miR-200 and/or miR141. This is an important experiment that would allow the relative contributions of the lncRNA and the miRNA be differentiated (which is a large part of the novelty and purpose of the paper).

Response: We thank the reviewer for the helpful comment. Considering the reviewer's suggestion, we've constructed a MIR200CHG mutation (MIR-Mut) that disrupts the interaction between miR-200 and MIR200CHG (**Figure R3A**). The mutant construct was transfected in Hs746T cells (**Figure R3B, new Supplementary Figure 9A**). qRT-PCR and western blotting showed that overexpression of MIR200CHG significantly increased the expression of E-cadherin and suppressed the expression of fibronectin, vimentin and ZEB1, and such variation was largely impaired by mutation in the miR-200c binding sites. (**Figure R3C, D, new Supplementary Figure 9B, C**). Wound healing assays revealed that miR-200c inhibition partially abrogated, while MIR200CHG mutant significantly abrogated the defects in cell migration caused by MIR200CHG overexpression in Hs746T cells (**Figure R3E, new Supplementary Figure 9D**). Overall, these findings suggest that MIR200CHG regulates EMT in a partially miR-200c-dependent manner and there are other potential target genes for MIR200CHG. We've added these results in the manuscript (Lines 321-324).

Figure R3. (A) The corresponding mutant form (MIR-Mut) with the predicted miR-200c binding site mutated is shown. **(B)** The expression of miR-200c was detected by qRT-PCR in Hs746t. **(C)** The mRNA expression of Fibronectin, E-Cadherin, Vimentin and Zeb1 were detected by qRT-PCR in Hs746t. **(D)** The protein expression of Fibronectin, E-Cadherin, Vimentin and Zeb1 were detected by western blot in Hs746t. **(E)** Wound healing analysis of Hs746T.

4. Code used in this study should be made available with the paper (e.g., through Github). This is clearly preferable to 'available upon request'

Response: We thank the reviewer for the suggestion. We've packed and uploaded the code to Github (<https://github.com/CityUHK-CompBio/GC-MIR200CHG>).

5. Data in Figure 3E-G is not particularly convincing – I agree that the correlations are apparent, but the text overstates the potential significance of methylation. Suggest a more balanced description of these data (or adding additional functional experiments).

Response: We are grateful for the important comment. As suggested, we have revised the description of these data (Lines 156-157). In addition, we have added additional functional experiments. The de-methylation treatment of two MSS/EMT cell lines (SNU668 and Hs746T) by 5-Aza resulted in significant decrease of cell migration and invasion (**Figure R4A, B, new Figure 3G, H**).

Figure R4. (A) Wound healing analysis of Hs746T and SNU668 cells treated with 5-AZA or DMSO. Scale bar, 200 μm. **(B)** Transwell chamber analysis of Hs746T and SNU668 cells treated with 5-AZA or DMSO. Scale bar, 50 μm.

6. Repeating some of the experiments in Figures 4 and 5 with a MIR200CHG variant that disrupts each (and both) of the two miRNAs would greatly strengthen the conclusions made in the paper.

Response: As suggested, we have constructed a MIR200CHG (MIR-Mut) that disrupts the interaction between miR-200c and MIR200CHG and repeated the experiments in the **Figure R3** and **new Supplementary Figure 9A-D**.

Figure R3. (A) The corresponding mutant form (MIR-Mut) with the predicted miR-200c binding site mutated is shown. **(B)** The expression of miR-200c was detected by qRT-PCR in Hs746T. **(C)** The mRNA expression of Fibronectin, E-Cadherin, Vimentin and Zeb1 were detected by qRT-PCR in Hs746T. **(D)** The protein expression of Fibronectin, E-Cadherin, Vimentin and Zeb1 were detected by western blot in Hs746T. **(E)** Wound healing analysis of Hs746T.

7. The apparent pairing between the lncRNA and miR-200 is rather minimal and less than what would be expected given the model – this could be directly tested by mutating the binding site.

Response: We appreciate the reviewer for another helpful suggestion. We constructed MIR200CHG mutants with mutations of predicted miR-200c binding site (Mut-MS2), and observed that mutations completely abolished the interaction between MIR200CHG and miR-200c (**Figure R5A, new Figure 6I**). RNA stability assay showed that enforced expression of wild-type MIR200CHG, but not mutant MIR200CHG, significantly stabilized miR-200c (**Figure R5B, new Figure 6J**). Altogether, these data

demonstrated that MIR200CHG is directly binding to miR-200c and could stabilize miR-200c.

Figure R5. (A) The corresponding mutant form (Mut-MS2) with the predicted miR-200c binding site mutated is shown (top). HEK293T cells were transfected with negative control (Con-MS2), vectors containing wild-type (WT-MS2) or mutated (Mut-MS2) MIR200CHG followed by MS2-RIP assay. qRT-PCR analysis showed the interaction of miR200CHG with miR-200c in HEK293T cells transfected with Con-MS2, WT-MS2 or mutated Mut-MS2. **(B)** qRT-PCR analysis showed the extending miR-200c half-life by overexpressing wild-type MIR200CHG but not mutant MIR200CHG.

Minor

1. There are publications linking lncRNAs to miRNAs via TDMD – these should be cited in the introduction.

Response: As suggested, we have added corresponding information in the introduction as follows (Lines 43-47).

A few lncRNAs have been reported to direct miRNA degradation through TDMD. For example, the lncRNA Cyrano uses an extensively paired site to miR-7 to trigger destruction of miR-7¹ (PMID: 29887379). The near-perfect miRNA binding site located in the lncRNA libra in zebrafish selectively triggers miR-29b destabilization through 3' trimming² (PMID: 29483647). Yet it remains elusive whether lncRNAs could stabilize miRNAs in TDMD.

2. The authors should explicitly state that the initial analyses use transcriptomic data (as they introduce the approach).

Response: We agreed and have explicitly stated that the initial analyses were based on transcriptomic data in the result part as follows (Lines 66-68).

To investigate whether the four GC subtypes differ in biology, as reported previously, we performed comprehensive functional characterizations on three independent GC cohorts (TCGA, ACRG and GSE15459) with transcriptomic data.

3. Line 160, remove 'strongly'

Response: Agreed and removed in the revised manuscript (Line 174).

4. Line 219, change 'synergic/cooperative' to 'as a consequence of MIR200CHG transcription' None of the data (in this section) speaks to cooperativity nor synergism.

Response: We thank the reviewer for the suggestion, and have deleted 'synergic/cooperative' accordingly (Lines 237-238).

5. Figure 7A – do the '...' in ZEB1 indicate missing nucleotides, if so, how many?

Response: We apologize for overlooking the length of the gaps during complementary pairing between ZEB1 and miR-200c and thank the reviewer for pointing out the issue. There was indeed a huge gap of 3007 nucleotides, as represented by '...', which was not an optimized pairing result. To improve the predictive analysis, in the revision we employed TargetScan, which incorporates a novel algorithm called TDMDScore to systematically scan and evaluate potential miRNA-target interactions that induce TDMD. A higher TDMDScore indicates a greater likelihood of TDMD being triggered by the interaction. Our findings reveal multiple binding sites of miR-200c on ZEB1 (**Figure R6A**). The binding site with the highest TDMDScore was selected and shown in Figure R6B. These results strongly suggest that ZEB1 is a reliable target for TDMD induced by miR-200c.

Figure R6. (A) Predicted binding sites of miR-200c on ZEB1 3'UTR by TargetScan. **(B)** The complementary pairing pattern between ZEB1 and miR-200c which may trigger TDMD.

6. Are other miR-200 paralogs expressed in these cells? If so, how does the model accommodate such expression?

Response: We thank the reviewer for the insightful comment. We've checked the expression of other miR-200 paralogs including miR-200a, miR-200b and miR-429 in the TCGA and CCLE dataset (**Figure R7A**). They all showed significantly lower expression in the MSS/EMT subtype. We used lncTAR and miRanda to simultaneously predict the binding sites of MIR200CHG and miR-200 paralogs. The results showed that MIR200CHG required the lowest binding free energy to bind to miR-200c and miR-429 (**Figure R7B, C**). MS2-RIP was performed to pull down endogenous RNAs and proteins associated with MIR200CHG. The results showed that miR-200c and miR-429 were significantly associated with MIR200CHG (**Figure R7D, E**). The data indicates that miR-200c and miR-429 but not other three miR-200 paralogs, have the ability to bind to and be stabilized by MIR200CHG.

Figure R7. (A) Boxplots showed significantly differential expression of miR-200a, miR-200b, and miR-429 between the MSS/EMT subtype and non-MSS/EMT subtypes in the TCGA cohort and CCLE cohort. **(B)** lncTAR showed predicted binding sites of

miR-200 paralogs on MIR200CHG. LncTAR utilized a variation on the standard "sliding" algorithm approach to calculate the binding free energy (ΔG) and normalized binding free energy (ΔG) to find the minimum free energy joint structure. **(C)** miRanda showed predicted binding sites of miR-200 paralogs on MIR200CHG. The miRanda score and binding free energy of the miRNA and MIR200CHG were shown. **(D)** Schematic diagram (top) and the corresponding mutant form (Mut-MS2) with the predicted miR-200c binding site mutated is shown (bottom). **(E)** MS2-RIP and qRT-PCR analyses showed the interaction of MIR200CHG with miR-200 paralogs in HEK293T cells.

Reviewer #2 (Remarks to the Author)

The association of MIR200CHG has been reported in multiple cancer types (renal cell carcinoma, urothelial cancer, breast cancer). Its role seems to have varied. For example, in bladder cancer, OE is associated with poor prognosis (contradicts your results), similarly, no expression produces poor outcome in breast cancer (similar to your results). Prior publications have stated that MIR200CHG may be an oncoMiR or TSGMiR depending on the context. This issue is serious and needs to be addressed.

Response: We thank the reviewer for the important insight, and we have added our discussion to the revised manuscript (Lines 393-406).

It's often the case that a lncRNA could be an oncogene in one cancer type yet as a tumor suppressor in other cancers, such as MALAT1³ (PMID: 32174966). MALAT1 was reported to be upregulated in human cancers such as renal cell carcinomas and bladder cancer, and could induce cancer cell proliferation, survival, migration, invasion, and metastasis. In contrast, MALAT1 has very recently been found to be downregulated in colorectal and breast cancers, and low MALAT1 expression was associated with poor survival of patients.

Indeed, MIR200CHG (also known as U47924.27) has been investigated in several other malignancies. Tang et al. reported that MIR200CHG was highly expressed in breast cancer tissues and could promote breast cancer proliferation, invasion, and drug resistance by interacting with and stabilizing YB-1⁴ (PMID: 34272387), which seems to be contradictory to our conclusion. However, the expression of MIR200CHG was also found to be heterogeneous in breast cancer cell lines in this study. The TNBC cell line MDA-MB-231, which is mesenchymal stem-like and more aggressive, showed little expression of MIR200CHG compared to luminal A cell lines such as MCF7 and T47D. The observation of heterogeneity in the expression of MIR200CHG is highly consistent with our findings in gastric cancer.

In addition, MIR200CHG has been reported to be a prognostic biomarker by serving as a protective factor or risk factor in bladder urothelial carcinoma⁵ (PMID: 30359990), lung adenocarcinoma⁶ (PMID: 35422758), melanoma⁷ (PMID: 28225791), and

colorectal cancer⁸ (PMID: 35664303) based on *in silico* analysis of public datasets. Therefore, we agreed with the reviewer that the function of MIR200CHG may be cancer tissue-dependent, cancer subtype-dependent and impacted by other factors such as subcellular localization. The biological functions and regulatory mechanisms of MIR200CHG rely on more in-depth investigations in a specific context.

Your data shows that MIR200CHG is down regulated (poor prognosis) in gastric cancer. Once an LNC has low expression or loss of expression, it would be difficult to target therapeutically, yet you did not seriously pursue the two that are upregulated in gastric cancer (AC104083.1 and LINC00578). Pursuing these could be interesting and potential worthy of pursuit in the clinic.

Response: We agree with the reviewer that it could be more difficult to target therapeutically for lowly expressed molecules. However, it has been reported that nanoparticle-mediated delivery could be a choice in the clinic⁹ (PMID: 35986402). Moreover, we believe the methylation inhibitor for MIR200CHG promoter is also a possible choice as indicated by the functional experiments (**new Figure 3G, H**).

We agree that these lncRNAs (AC104083.1 and LINC00578) are worth pursuing even if they showed no significant clinical association with survival in the TCGA dataset. We've initiated the study of these two lncRNAs.

You emphasized heterogeneity of gastric cancers but following the MIR200CHG research trajectory, heterogeneity concept is lost. MSS/EMT subtype was chosen but one is sure it is a heterogeneous subtype. Mesenchymal subtype have been described more than ones. It is unclear if there is a subtype associated with OE of MIR200CHG and another one with its low expression. Overall, agree that MIR200CHG can serve as a prognostic marker (how many more do we need in gastric cancer) but if this can not be further pursued then what to become of it.

Response: We thank the reviewer for the comments. We started from the molecular heterogeneity of gastric cancer and discovered a distinct lncRNA expression pattern in the MSS/EMT subtype, which is associated with poor prognosis. Based on a systematic network inference analysis, we identified MIR200CHG as a master regulator of EMT specifically underlying the MSS/EMT subtype, which is our major interest in the study.

The mesenchymal subtype exists in many cancers and was recently found to be heterogeneous in cancers such as glioblastoma¹⁰ (PMID: 33566263). However, as far as we known, there is no established subtyping system for reference in gastric cancer, which could further subdivide the mesenchymal subtype. In our analysis, both the violin plot (**Figure R8**) and heatmap (**Figure 1A**) showed that the MSS/EMT patients had consistently and significantly lower expression levels of MIR200CHG than the non-MSS/EMT patients in the TCGA dataset ($P = 1.79e-10$). Moreover, there was no expression of MIR200CHG in the six GC MSS/EMT cell lines in the CCLE database

(**new Supplementary Figure 4C**) and little expression of MIR200CHG in the three GC MSS/EMT cell lines in our *in vitro* experiments (**Figure 3D**).

Nevertheless, this suggestion is indeed very constructive, and we are dedicated to taking it into consideration in our future work.

Figure R8. Violin plot shows the MSS/EMT patients in the TCGA dataset had significantly lower expression levels of MIR200CHG than the non-MSS/EMT patients.

There is little mention of published literature on MIR200CHG in cancers, and discordant results should be explained by generating a pan-cancer integrated signature that might shed some light on its function in various contexts.

Response: We thank the reviewer for the helpful suggestion, which shed light on our future work. In the revised manuscript, we have made a detailed literature review about MIR200CHG in different cancers in the discussion (Lines 393-406). However, in this study we focused more on the subtype-specific functional role and regulatory mechanism of MIR200CHG in gastric cancer. Our pan-cancer analysis showed largely consistent results across many different cancers, suggesting that the function of MIR200CHG in inhibiting the EMT pathway may represent a general mechanism to suppress cancer metastasis.

Was MIR200CHG explored in other gastric cancers (other than MSS/EMT), this is not clear.

Response: We thank the reviewer for the comment. We've made a thorough literature review about MIR200CHG (U47924.27). However, MIR200CHG has not been explored in gastric cancers before.

So ultimately, we have low expression of MIR200CHG. Various functional experiments to document that hypermethylation (which is often the case for TSGs in cancer in general, some exceptions noted), EMT can be reversed if MIR200CHG is induced, the

lymph node metastases model is suboptimal, and its binding and stabilization of miR-200c, and its prognostic value. The pan-cancer analysis is sub optimally described and contractions exist and does not address published literature.

Multivariate analysis is left out.

Response: We thank the reviewer for the useful suggestion. We've added the multivariate analysis as below (**new Table 2**). MIR200CHG was demonstrated to be an independent prognostic factor.

Features	beta	HR	HR (95% CI)	P-value
Gender: M vs. F	0.065	1.07	1.07 (0.61-1.87)	0.8200
T stage: T34 vs. T12	0.830	2.30	2.3 (1.31-4.02)	0.0035
M stage: M1 vs. M0	0.930	2.53	2.53 (1.13-5.64)	0.0240
MIR200CHG	-0.410	0.66	0.66 (0.48-0.93)	0.0160
AC104083.1	0.051	1.05	1.05 (0.74-1.50)	0.7800
LINC00578	-0.150	0.86	0.86 (0.52-1.40)	0.5400

No new targets and no drug

Response: We thank the reviewer for the suggestion on our future work. Our major objective in this study is to elucidate the subtype-specific regulatory mechanism of MIR200CHG in gastric cancer. Development and validation of novel therapeutics will be our long-term goal in the future.

No human gastric cancer models

Response: We apologize for the lack of a human gastric cancer model. Currently, only human MIR200CHG expression has been found without corresponding murine MIR200CHG. Therefore, all gain and loss of function experiments were conducted in cell lines derived from gastric cancer patients. For the mouse lymph node metastasis model, we used tumor cell lines derived from human gastric cancer patients which have also been utilized in several studies on gastric cancer. In response to the reviewer's suggestion, we further established a mouse peritoneal metastasis model commonly used to study gastric cancer metastasis. Our data showed that inhibition of MIR200CHG aggravated peritoneal metastasis (**Figure R9 A, B, new figure 5G, H**) and decreased overall survival rate in mice (**Figure R9D, new figure 5J**). Inhibition of MIR200CHG led to faster weight gain in mice reflecting the rate of ascites production (**Figure R9C, new figure 5I**). Additionally, we are attempting to collect human gastric cancer tissues and construct PDX models. We've added these results in the manuscript (Line 218-223).

Figure R9. (A) Representative images of the mouse primary gastric tumor and peritoneal metastasis established with NCI-N87 Scramble and MIR200CHG-knockdown cell lines (n = 10). **(B-C)** Number of peritoneal metastasis nodules and body weight of nude mice. **(D)** Survival of nude mice with NCI-N87 Scramble and MIR200CHG-knockdown tumors. **(E)** HE, Immunocytochemical analysis of E-cadherin and vimentin, and RNA FISH of MIR200CHG of the primary tumours and peritoneal metastasis of NCI-N87 Scramble and MIR200CHG-knockdown cell lines. Black scale bar, 100 μ m. White scale bar, 20 μ m.

Reviewer #3 (Remarks to the Author):

In the current manuscript, the authors investigated the subtype-specific expression and general function of the noncoding RNA MIR200CHG in gastric cancer (GC). This ncRNA is significantly repressed in the MSS/EMT subtype of GC which seems to be due to epigenetic silencing, i.e. promoter hypermethylation. Overexpression of MIR200CHG inhibited migration and invasion of GC cells of the MSS/EMT subtype in vitro and metastasis in vivo whereas knockdown of the ncRNA in GC cells of other subtypes induced an epithelial-mesenchymal transition and enhanced cell motility. Mechanistically, the authors show that the spliced MIR200CHG is able to bind to the intronically encoded mature miR-200c, but not to miR-141 that is also derived from the same intron. The direct RNA:RNA interaction seems to stabilize miR-200c by

preventing it from binding to its downstream mRNA targets. In particular, the interaction between miR-200c and Zeb1 mRNA is reduced upon MIR200CHG overexpression. This in turn prevents the target-dependent miR-200c decay.

Overall, the study follows a logical flow and the manuscript is well-written. The authors assembled lots of data and included several important controls in their assays. Therefore, I only have some minor concerns that should be addressed:

1) While most assays contain relevant and important controls some do not. For example, all pulldown assays lack a proper specificity control, i.e. a non-binding negative control. For instance, Figure 7C shows the reduced binding of Zeb1 to Ago2 upon MIR200CHG overexpression as expected. The authors should show the enrichment of other mRNAs, either with or without miR-200c binding sites. Furthermore, RNA FISH assays (Suppl. Figure 4A) lack negative controls, i.e. cells that do not express MIR200CHG.

Response: We thank the reviewer for the constructive comments. We have repeated most of the RNA-pulldown and RIP, and have added GAPDH and U6 as negative controls. We also have repeated the RNA-FISH using Hs746T/Vector as a negative control in **supplementary Figure S5A (Figure R10)**.

Figure R10. Representative image for MIR200CHG localization in NCI-N87 and Hs746T cells. Scale bar, 100 μ m.

2) The RNA FISH analysis (Suppl. Figure 4A) suggests that MIR200CHG localizes in cytoplasmic speckles. Do these speckles contain Ago2? Please show co-localization. Do the authors have any idea what the nature of these speckles might be? Stress granules?

Response: We thank the reviewer for the insightful suggestion. miRNAs are transcribed as precursor molecules, which are subsequently cleaved by endoribonucleases Drosha and Dicer. Mature miRNAs bind to a member of the Argonaute (AGO) protein family to form the RNA-induced silencing complex (RISC). Previous studies have shown that AGO2 is the core component of RISC and regulates mRNA degradation. Through FISH-IF experiments, we found partial co-localization between MIR200CHG and AGO2, as well as between MIR200CHG and DICER -

another marker for RISC. We also used G3BP1 as a marker of stress granules and detected whether MIR200CHG was located in stress granules. The results showed that MIR200CHG was partially co-located with G3BP1, but not in the cytoplasmic speckles (**Figure R11, new Supplementary Figure 7K**). These results suggested that the speckles might be RISC. We've added these results to the revised manuscript (Lines 269-273).

Figure R11. IF-FISH demonstrated the co-localization of MIR200CHG and AGO2, DICER or G3BP1. Scale bar, 100 μ m.

3) In Figure 6F, the authors should include an additional control and repeat the assay with miR-141 mimics as well.

Response: We thank the reviewer for the suggestion. We have used GAPDH as negative control and repeat the assay with miR-141 mimics in **Figure 6F (Figure R12)**. The result showed that MIR200CHG was specifically enriched in an anti-AGO2 antibody-associated complex with miR-200C but not miR-141.

Figure R12. AGO2 RIP assay showed that both MIR200CHG and miR-200c but not miR-141 were bound to AGO2.

4) In Figure 6C (and also in other stability assays shown in the Supplementary Data), the half-life of the RNA of interest (here: miR-200c) varies dramatically between a both controls (vector control, shScramble). The changes are within the range of the

experimental groups. Any explanation for this variation? In theory, the half-life should not be altered by the controls.

Response: We thank the reviewer for the helpful comments. MSS/EMT cell line - Hs746t was used for vector control, which had low expression of miR-200c. However, non-MSS/EMT cell line - NCI-N87 was used for shScramble, which showed high expression of miR-200c (**Figure R13, new Supplementary Figure 7C**). The differential expression of miR-200c in these two cell lines may be the cause of the variation of half-life.

Figure R13. qRT-PCR analysis of miR-200c expression in the MSS/EMT and non-MSS/EMT subtypes GC cell lines.

5) Does the modulation of Zeb1 expression affect miR-200c stability?

Response: We are grateful for this constructive comment. We have used siRNA to inhibit ZEB1 expression in Hs746t cells and detected the half-life of miR-200c. The results showed that ZEB1 knockdown increased the half-life of miR-200c (**Figure R14, new Supplementary Figure 8D**), indicating that ZEB1 plays a critical role in regulating the stability of miR-200c in GC cells. We've added this result in the revised manuscript (Lines 286-287).

Figure R14. The knockdown of ZEB1 extends the RNA half-life of miR-200c.

6) It is unclear how the miR-S probe depicted in Suppl. Figure 6J, which is complementary to the MIR200CHG but not miR-200c, should be able to bind to endogenous miR-200c as stated in line 244ff and shown in Figure 6G and 6H. Please describe this assay in more detail and check the sequence and/or wording.

Response: We thank the reviewer for the suggestion. To further determine whether MIR200CHG directly binds to miR-200c, we constructed a short probe of MIR200CHG with a length of 24bp, which contains the predicted binding site to miR-200c (MIR-S). The short probe MIR-S was co-incubated with total RNA from NCI-N87 cell lysates (**Figure 6G**) or miR-200c mimic (**Figure 6H**), and then RNA pulldown and qRT-PCR were performed. The interaction of miR-200c and MIR-S probe confirmed the reliability of predicted miR-200c binding sites of MIR200CHG. We've made revision in the revised manuscript (Lines 257-265).

7) Please provide a more detailed description of the miRNA quantification (stem-loop RT-PCR protocol).

Response: We thank the reviewer for the comment. Total RNA was isolated from gastric cancer cell lines using Trizol reagent (Invitrogen, USA) following the manufacturer's protocol. Target-specific cDNA synthesis proceeds with the stem-loop RT primer and then qPCR subsequently proceeds with the forward primer to rapidly achieve a quantitative result. We synthesized cDNA by reverse transcription reaction using a miRNA 1st Strand cDNA Synthesis Kit (by stem-loop) (Vazyme #MR101-01, China). qPCR was conducted using a miRNA Universal SYBR qPCR Master Mix (Vazyme #MQ101, China) according to manufacturer's instructions. Results were normalized using U6 as an internal control. To account for the assessment of technical variability, the assays were performed in triplicate for each case. Stem-loop primers are designed by Vazyme miRNA Design V1.01. and sequences are shown in **Supplementary Table 6**. We have provided detailed description of the miRNA quantification in Methods (Lines 499-508).

References

- 1 Kleaveland, B., Shi, C. Y., Stefano, J. & Bartel, D. P. A network of noncoding regulatory RNAs acts in the mammalian brain. *Cell* **174**, 350-362. e317 (2018).
- 2 Bitetti, A. *et al.* MicroRNA degradation by a conserved target RNA regulates animal behavior. *Nature structural & molecular biology* **25**, 244-251 (2018).
- 3 Chen, Q., Zhu, C. & Jin, Y. The oncogenic and tumor suppressive functions of the long noncoding RNA MALAT1: an emerging controversy. *Frontiers in genetics* **11**, 93 (2020).
- 4 Tang, L. *et al.* Long non-coding RNA MIR200CHG promotes breast cancer proliferation, invasion, and drug resistance by interacting with and stabilizing YB-1. *NPJ breast cancer* **7**, 94 (2021).
- 5 He, R.-Q. *et al.* RNA-sequencing data reveal a prognostic four-lncRNA-based risk score for bladder urothelial carcinoma: an in silico update. *Cellular Physiology and Biochemistry* **50**, 1474-1495 (2018).
- 6 Mai, S. *et al.* Development and validation of lactate metabolism-related lncRNA signature as a prognostic model for lung adenocarcinoma. *Frontiers in Endocrinology* **13**, 829175 (2022).
- 7 Wang, S. *et al.* Characterization of long noncoding RNA and messenger RNA signatures in melanoma tumorigenesis and metastasis. *PloS one* **12**, e0172498 (2017).
- 8 Tang, Q. *et al.* Discovery and validation of a novel metastasis-related lncRNA prognostic signature for colorectal cancer. *Frontiers in Genetics* **13** (2022).
- 9 Mao, W. *et al.* Transfection with Plasmid-Encoding lncRNA-SLERCC nanoparticle-mediated delivery suppressed tumor progression in renal cell carcinoma. *Journal of Experimental & Clinical Cancer Research* **41**, 252 (2022).
- 10 Cifarelli, C. P., Jacques, A. & Bobko, A. Heterogeneity of radiation response in mesenchymal subtype glioblastoma: molecular profiling and reactive oxygen species generation. *Journal of Neuro-oncology* **152**, 245-255 (2021).

REVIEWERS' COMMENTS

Reviewer #1 (Remarks to the Author):

The authors have completed a substantial number of new experiments and analyses, resulting in a greatly improved manuscript. I recommend that the paper be accepted, although I believe Figures R2 and R7 (or at least some of R7) be included as part of the supplemental figures.

Reviewer #3 (Remarks to the Author):

The authors fully addressed my concerns. I do not have any additional questions.

Reviewer #4 (Replacing Reviewer #2, Remarks to the Author):

The authors all in all responded sufficiently to all questions raised. Only concerning question 3 (heterogeneity within the MSS/EMT subtype), the authors should add the presented violin plot from the rebuttal letter (Fig. R8) in a Supplementary Figure and add in the paper a sentence or two, that heterogeneity does exist also within the MSS/EMT subtype, and that the role of MIR200CHG in normal expressing MSS/EMT gastric cancers need more dedicated research.

Point by point response to reviewers' comments:

We sincerely appreciate the reviewers for the positive comments on our revised manuscript and further insights. We have addressed all the comments as follows and made revisions accordingly in the updated manuscript.

Reviewer #1 (Remarks to the Author):

The authors have completed a substantial number of new experiments and analyses, resulting in a greatly improved manuscript. I recommend that the paper be accepted, although I believe Figures R2 and R7 (or at least some of R7) be included as part of the supplemental figures.

Response: We sincerely thank the reviewer for the positive comments and further suggestions. In our revised manuscript, we have included Figures R2 (**new Supplementary Figure 7f, g, Line 254-259**), and R7 (**new Supplementary Figure 8, Line 297-306**).

Reviewer #3 (Remarks to the Author):

The authors fully addressed my concerns. I do not have any additional questions.

Response: We thank the reviewer for the previous comments, which were very helpful.

Reviewer #4 (Replacing Reviewer #2, Remarks to the Author):

The authors all in all responded sufficiently to all questions raised. Only concerning question 3 (heterogeneity within the MSS/EMT subtype), the authors should add the presented violin plot from the rebuttal letter (Fig. R8) in a Supplementary Figure and add in the paper a sentence or two, that heterogeneity does exist also within the MSS/EMT subtype, and that the role of MIR200CHG in normal expressing MSS/EMT gastric cancers need more dedicated research.

Response: We thank the reviewer for the important insight. In our revised manuscript, we have added the Violin plot (**new Supplementary Figure 12**) and added the corresponding text in 'Discussion' (**Line 435-438**).